# Generalizable Domain Adaptation for Sim-and-Real Policy Co-Training

**Shuo Cheng**[1*]     **Liqian Ma**[1*]     **Zhenyang Chen**[1]
**Ajay Mandlekar**[2†]   **Caelan Garrett**[2†]     **Danfei Xu**[1]
[1] Georgia Institute of Technology    [2] NVIDIA Corporation

[*] and [†] denote equal contribution

{shuocheng, mlq}@gatech.edu

## Abstract

Behavior cloning has shown promise for robot manipulation, but real-world demonstrations are costly to acquire at scale. While simulated data offers a scalable alternative, particularly with advances in automated demonstration generation, transferring policies to the real world is hampered by various simulation and real domain gaps. In this work, we propose a unified sim-and-real co-training framework for learning generalizable manipulation policies that primarily leverages simulation and only requires a few real-world demonstrations. Central to our approach is learning a domain-invariant, task-relevant feature space. Our key insight is that aligning the *joint distributions* of observations and their corresponding actions across domains provides a richer signal than aligning observations (marginals) alone. We achieve this by embedding an Optimal Transport (OT)-inspired loss within the co-training framework, and extend this to an Unbalanced OT framework to handle the imbalance between abundant simulation data and limited real-world examples. We validate our method on challenging manipulation tasks, showing it can leverage abundant simulation data to achieve up to a 30% improvement in the real-world success rate and even generalize to scenarios seen *only in simulation*. Project webpage: `https://ot-sim2real.github.io/`.

## 1   Introduction

Behavior cloning [1] is a promising approach for acquiring robot manipulation skills directly in the real world, due to its simplicity and effectiveness in mimicking expert demonstrations [2, 3]. However, achieving robust and generalizable performance requires collecting large-scale datasets [4, 5] across diverse environments, object configurations, and tasks. This data collection process is labor-intensive, time-consuming, and costly, posing significant challenges to scalability in real-world applications.

Recently, with rapid advancements in physics simulators [6, 7], procedural scene generation [8, 9], and motion synthesis techniques [10, 11], there has been growing interest in leveraging simulation as an alternative source of training data. These simulation-based approaches enable scalable and controllable data generation, allowing for diverse and abundant supervision at a fraction of the real-world cost. However, transferring policies trained in simulation to the physical world remains a non-trivial challenge due to sim-to-real gap—the discrepancies between the simulated and real-world environments that a policy encounters during execution. These differences can manifest in various forms, such as variations in visual appearance, sensor noise, and action dynamics [12, 13]. In particular, learning visuomotor control policies that remain robust under changing perceptual conditions during real-world deployment continues to be an open area of research.

Common strategies to bridge this domain gap include domain randomization [12, 13] and data augmentation [14, 15], though these often require careful tuning. Domain adaptation (DA) techniques

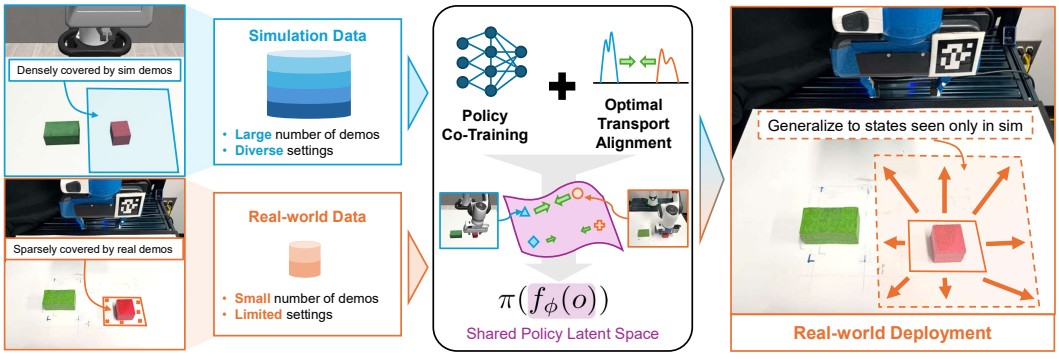

Figure 1: **Sim-and-Real Co-Training with Optimal Transport**. We use behavior cloning to train a real-world policy from sparse real-world and dense simulation demos. Leveraging Optimal Transport to align feature spaces, our method enables generalization to scenarios seen only in simulation.

aim to explicitly align distributions, either at pixel [16, 17] or feature levels [18, 19, 20]. However, many feature-level methods align only marginal observation distributions (e.g., MMD [18, 19]), which can be insufficient for fine-grained manipulation alignment as it may not preserve action-relevant relationships across domains. More recently, sim-and-real co-training—simply training a single policy on mixed data from both domains [21, 22]—has shown surprising effectiveness. We argue that while beneficial for data diversity, such co-training approaches typically lack explicit constraints for feature space alignment across domains, potentially hindering optimal transfer and generalization because they don't enforce a consistent mapping of task-relevant structures.

We present a unified sim-and-real co-training framework that explicitly learns a shared latent space where observations from simulation and the real world are aligned and preserve action-relevant information. Our key insight is that aligning the *joint distributions* of observations and their corresponding actions or task-relevant states across domains provides a direct signal for learning transferable features. Concretely, we leverage Optimal Transport (OT) [23] as an alignment objective to learn representations where the geometric relationships crucial for action prediction are consistent, irrespective of whether the input comes from simulation or the real world. Further more, to robustly handle the *data imbalance* in co-training with abundant simulation data and limited real-world data, we further extend to an Unbalanced OT (UOT) formulation [24, 25] and develop a temporally-aware sampling strategy to improve domain alignment learning in a mini-batch OT setting.

**Our contributions are**: (1) a sim-and-real co-training framework that learns a domain-invariant yet task-salient latent space to improve real-world performance with abundant simulation data, (2) an Unbalanced Optimal Transport framework and temporally-aware sampling strategy to mitigate data imbalance and improve alignment quality in mini-batch OT training, (3) comprehensive experiments using both image and point-cloud modalities, evaluating sim-to-sim and sim-to-real transfer across diverse manipulation tasks, demonstrating up to a 30% average success rate improvement and achieving generalization to real-world scenarios for which the training data only appears in simulation.

## 2 Related Work

**Behavior Cloning for Robot Manipulation.** Behavior cloning (BC) trains policies to map observations to actions by imitating expert demonstrations [2, 26, 27, 28], offering an effective path to human-like manipulation skills. Generalization heavily depends on dataset diversity. While some efforts focus on large-scale real-world data collection [5, 4] or more efficient collection techniques [29, 30, 31], this remains costly and time-consuming. Physical simulators provide a low-cost alternative, with automatic motion synthesis leveraging privileged information to generate large-scale simulated demonstrations [10, 32, 33]. Our work combines abundant simulated data with few real-world demonstrations to train robust BC policies.

**Sim-to-real Transfer and Co-training.** Policies trained solely in simulation often underperform in the real world due to the sim-to-real gap—discrepancies in visual appearance and dynamics. For quasi-static manipulation, the visual domain gap is typically the primary bottleneck. Domain

randomization exposes policies to varied simulated visual conditions to build robustness to real-world variability [13, 34, 35]. However, its success depends on how well randomized parameters cover true real-world distributions, often requiring manual tuning. Domain adaptation (DA) explicitly aligns source (simulation) and target (real) domains [36, 37]. Pixel-level DA uses image translation to make simulated images resemble real ones [16, 17, 38]. Feature-level DA, which is often more scalable for end-to-end learning, focuses on learning domain-invariant representations [19, 20, 18, 39, 40]. Sim-and-real co-training, where a policy is jointly trained on mixed data [21, 22], offers a simple and effective alternative. While co-training enhances generalization through data diversity, it typically lacks explicit constraints to align learned feature spaces across domains. We build upon co-training by incorporating feature-level domain adaptation via Optimal Transport to promote latent space alignment, thereby improving real-world policy performance.

**Optimal Transport for Domain Adaptation.** Optimal Transport (OT) offers a principled framework for aligning distributions, widely adopted for domain adaptation [23, 41, 42, 43, 44, 45, 46]. Traditional OT methods compute a transport plan between source and target samples, then train a new model on the transported source. Most relevant to us is DeepJDOT [47], which builds on JDOT [44] to align joint distributions of features and (pseudo) labels for unsupervised domain adaptation, where target labels are unavailable. Our work builds on these principles for sim-and-real co-training imitation. Unlike unsupervised DA, we leverage available action or state labels from limited real-world demonstrations as "soft" supervision to guide a more task-relevant alignment of joint observation-label distributions across domains. To robustly handle the inherent data imbalance between abundant simulation and scarce real data, we incorporate an Unbalanced OT (UOT) loss [24] into our co-training framework and develop a temporally-aware sampling strategy to improve mini-batch UOT training.

# 3 Preliminaries and Problem Setting

Our method builds on the principle of Optimal Transport (OT) for aligning two empirical distributions. Let $\mathbf{U} = \{u_i\}_{i=1}^n$ and $\mathbf{V} = \{v_j\}_{j=1}^m$ represent the data points drawn from a *source* domain and a *target* domain, with corresponding empirical distributions $\mathbf{p} = \sum_{i=1}^n p_i \delta_{u_i}$ and $\mathbf{q} = \sum_{j=1}^m q_j \delta_{v_j}$. We define the ground cost matrix $C = (C_{i,j}) \in \mathbb{R}^{n \times m}$ with $C_{i,j} = c(u_i, v_j)$, where $c(\cdot, \cdot)$ is a cost function, which is often defined as squared Euclidean distance. Optimal Transport (OT) seeks to find an optimal plan $\Pi$ that maps the distribution $\mathbf{p}$ to $\mathbf{q}$ that minimizes the displacement cost $W_c(\mathbf{p}, \mathbf{q})$:

$$W_c(\mathbf{p}, \mathbf{q}) = \min_{\Pi \in \mathbb{R}_+^{n \times m}} \langle \Pi, C \rangle_F, \text{ s.t. } \Pi \mathbf{1}_m = \mathbf{p}, \Pi^\top \mathbf{1}_n = \mathbf{q}. \tag{1}$$

## 3.1 Problem Setting: Sim-and-Real Policy Co-Training

We address the challenge of learning robust real-world robotic manipulation policies $\pi$. Our approach minimizes the need for extensive real-world data collection by primarily leveraging abundant simulation data alongside a small set of real-world demonstrations. This is framed as a sim-and-real co-training problem [22, 48], where a single policy is trained on data from both domains. Specifically, we consider a source domain (simulation, denoted $src$) and a target domain (real-world, denoted $tgt$). We model the domains as Partially Observable Markov Decision Processes (POMDPs) that share an underlying, generally unobserved, state space $\mathcal{S}$ and an action space $\mathcal{A}$. The policy receives observations comprising high-dimensional visual input $o \in \mathcal{O}$ (e.g., RGB images, 3D point clouds) generated by the emission function $E : \mathcal{S} \mapsto \mathcal{O}$, together with low-dimensional proprioceptive information $x \in \mathcal{X}$ (e.g., robot joint angles, end-effector pose).

**Domain Gaps.** The central challenge is the domain gap, particularly the *visual observation gap*. For the same underlying robot and environment state $s \in \mathcal{S}$, visual observations emitted in simulation, $o_{src} = E_{src}(s)$, can differ significantly from those in the real world, $o_{tgt} = E_{tgt}(s)$. This discrepancy arises from factors like variations in visual appearance (textures, lighting), sensor noise, and rendering artifacts (e.g., differences between simulated ray casting and real-world light transport). As a result, the marginal observation distributions differ between the domains, namely $P_{src}(o_{src}) \neq P_{tgt}(o_{tgt})$. In contrast, actions $a$ and proprioceptive states $x$ are assumed to be largely consistent for a given $s$ due to consistent data generation strategies (discussed next) and accurate robot state estimation. While differences in dynamics also contribute to the domain gaps, our focus on learning quasi-static prehensile manipulation tasks from human-sourced demonstrations means that the dynamics gap is typically less dominant than the observation gap.

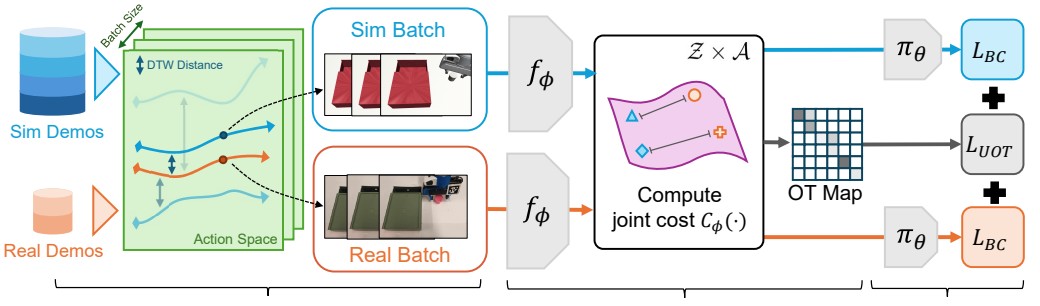

Figure 2: **Method Overview.** Our sim-and-real co-training framework learns a domain-invariant latent space to improve real-world performance using abundant simulation demos and a small number of real-world demos. It leverages an Unbalanced Optimal Transport loss and a temporal sampling strategy to address data imbalance and improve alignment quality during mini-batch training.

**Data Sources and Data Imbalance.** In the source (simulation) domain, we leverage the ability to automatically generate a large dataset of $N_{src}$ trajectories, $D_{src} = \{(o^i_{src}, x^i_{src}, a^i_{src})\}^{N_{src}}_{i=1}$. Specifically, we leverage an automated demonstration generation tool MimicGen [10], which utilizes privileged information available in simulation to create a diverse set of experiences covering a broad underlying state space, $S_{src}$. MimicGen populates synthetic demonstrations based on a handful of human demonstrations, which ensures that the generated data is behaviorally consistent with human demonstrations. We collect a limited number of $N_{tgt}$ demonstrations, $D_{tgt} = \{(o^j_{tgt}, x^j_{tgt}, a^j_{tgt})\}^{N_{tgt}}_{j=1}$ (where $N_{src} \gg N_{tgt}$), typically through human teleoperation. These real-world demonstrations will naturally cover a much smaller and potentially distinct subset of states, $S_{tgt}$. This difference in data coverage leads to the challenge of **partial data overlap**. While we assume there is a region of states common to both $S_{src}$ and $S_{tgt}$ where direct alignment is possible, a significant portion of $S_{src}$ (our rich simulated data) will not have corresponding real-world demonstrations in $S_{tgt}$. Conversely, $S_{tgt}$ might contain details specific to real world (e.g., demonstration behaviors). Effectively leveraging the entirety of $D_{src}$ for real-world performance, especially for states outside the direct sim-real overlap in demonstrations, is the problem we address.

**Objective: Generalizable Domain Adaptation.** Our goal is to learn a single, generalizable policy $\pi_\theta(a|z, x)$ and an observation encoder $f_\phi : \mathcal{O}_{src} \cup \mathcal{O}_{tgt} \rightarrow \mathcal{Z}$. This encoder maps high-dimensional visual observations $o$ from both source and target domains to a shared latent space $\mathcal{Z}$. The primary objective is to achieve high policy performance in the target (real-world) domain, especially in scenarios not explicitly covered by the limited real-world demonstrations $D_{tgt}$. This entails **two objectives**. First, for states within the overlapping regions of $S_{src}$ and $S_{tgt}$, we aim to learn high-quality embeddings $z = f_\phi(o)$ that are well-aligned across domains, such that $f_\phi(o_{src}) \approx f_\phi(o_{tgt})$ in corresponding states, facilitating effective policy learning. This is also the assumption of most co-training methods [21, 22]. Second, for states covered in $S_{src}$ but not in $S_{tgt}$, the encoder $f_\phi$ must produce embeddings $f_\phi(o_{tgt})$ for novel target observations that are consistent with the embeddings of their simulated counterparts $f_\phi(o_{src})$. This requires the learned representations to capture domain-invariant, task-relevant features, enabling policy to *generalize to target (real-world) scenarios for which training data is only present in the source (simulated) domain.*

## 4 Method

To learn a generalizable policy $\pi_\theta$ with robust feature $f_\phi$ from imbalanced sim-and-real datasets ($|D_{src}| \gg |D_{tgt}|$), as outlined in Section 3, we propose a co-training framework that explicitly aligns the latent representation through Optimal Transport (OT). Our core strategy is to leverage action information to guide the alignment of learned visual latent features ($z = f_\phi(o)$). This alignment objective simultaneously encourages the encoder $f_\phi$ to discover domain-invariant representations while preserving detailed information for action prediction. Specifically, we propose a formulation based on OT to align joint observation-action distributions (Sec. 4.1). We further address the data imbalance problem through Unbalanced Optimal Transport (UOT) (Sec. 4.2) and a temporally-aware sampling strategy (Sec. 4.3), all integrated into a unified co-training framework (Sec. 4.4).

## 4.1 Optimal Transport for Action-Aware Feature Alignment

While standard co-training methods [21, 22] offer implicit feature alignment, and marginal distribution matching (e.g., Maximum Mean Discrepancy [18]) can overlook fine-grained correspondences, we seek a more structured approach. To learn domain-invariant visual features $z = f_\phi(o)$ that are also predictive of actions $a \in \mathcal{A}$, we propose to align the joint distributions $P_{src}(f_\phi(o_{src}), a_{src})$ and $P_{tgt}(f_\phi(o_{tgt}), a_{tgt})$ using Optimal Transport. This encourages the encoder $f_\phi$ to learn representations where the geometric relationships between (visual feature, action) pairs are preserved across domains. By minimizing an OT-based loss, the encoder $f_\phi$ is trained to shape the embedding space $\mathcal{Z}$ such that structures relevant to action prediction are consistent between simulation and the real world.

Formally, given source samples $\{(o_{src}^i, a_{src}^i)\}_{i=1}^{N_{src}}$ and target samples $\{(o_{tgt}^j, a_{tgt}^j)\}_{j=1}^{N_{tgt}}$, we aim to find an optimal transport plan $\Pi^*$ and an optimal encoder $f_\phi^*$ that minimize the transportation cost between their joint distributions in the $(z, a)$ space. Based on the general OT formulation in Eq. 1, learning objective for the encoder $f_\phi$, and implicitly the transport plan $\Pi$, can be expressed as finding $f_\phi$ that minimizes the Wasserstein distance between $P_{src}(f_\phi(o_{src}), a_{src})$ and $P_{tgt}(f_\phi(o_{tgt}), a_{tgt})$: $\min_{f_\phi} W_C \left( P_{src}(f_\phi(o_{src}), a_{src}), P_{tgt}(f_\phi(o_{tgt}), a_{tgt}) \right)$. The ground cost $c$ ideally combine distances in both the learned visual latent space $\mathcal{Z}$ and the action space $\mathcal{A}$, for instance:

$$C_\phi \left( (f_\phi(o_{src}^i), a_{src}^i), (f_\phi(o_{tgt}^j), a_{tgt}^j) \right) = \alpha_1 \cdot d_\mathcal{Z}(f_\phi(o_{src}^i), f_\phi(o_{tgt}^j)) + \alpha_2 \cdot d_\mathcal{A}(a_{src}^i, a_{tgt}^j). \quad (2)$$

Minimizing this objective using an iterative algorithm like Sinkhorn [49] creates a bi-level optimization. In the inner loop, for a fixed $f_\phi$, an approximately optimal transport plan $\Pi$ is computed. In the outer loop, $f_\phi$ is updated to reduce the cost incurred by this plan. This process effectively trains the encoder $f_\phi$ to produce embeddings $z$ that make the source and target joint distributions $(z, a)$ less costly to align. A key advantage of OT is its ability to preserve geometric structures; by guiding alignment with action similarity (via $d_\mathcal{A}$), we shape the embedding function $f_\phi$ to cluster visual observations that lead to similar actions, irrespective of their domain of origin.

**Practical Implementation: Proprioception as Guidance.** While direct alignment of $(z, a)$ is principled, discrepancies in controller characteristics or action representations between simulation and real-world teleoperation can make $d_\mathcal{A}(a_{src}, a_{tgt})$ an unreliable indicator of behavioral similarity. As a robust practical compromise, we leverage proprioceptive information $x \in \mathcal{X}$ (e.g., end-effector pose), which is more consistently represented across domains (Section 3.1) and highly correlated with robot behavior. Thus, our implemented ground cost $c$ replaces actions $a$ with proprioceptive states $x$, which is used in our UOT formulation (detailed in Section 4.2).

## 4.2 Unbalanced Optimal Transport for Robust Alignment

The standard OT formulation (Eq. 1) enforces strict marginal constraints, requiring all mass from the source distribution to be transported to the target and vice-versa. This is problematic in our sim-to-real setting primarily due to significant data imbalance ($|D_{src}| \gg |D_{tgt}|$) and the partial overlap between the state spaces covered by $D_{src}$ and $D_{tgt}$ (Section 3.1). Standard OT would either distort the latent space by forcing many-to-few mappings or create spurious alignments between non-corresponding states. To address these challenges, we employ Unbalanced Optimal Transport (UOT) [25, 24]. UOT relaxes the hard marginal constraints of OT by introducing regularization terms that penalize deviations, thereby allowing for partial mass transport. This enables UOT to selectively align subsets of the distributions that are most similar according to the ground cost, while effectively down-weighting or ignoring the transport for dissimilar or unmatched portions.

**UOT Loss Formulation.** Consider a mini-batch of $N_{batch}$ source samples $\{(o_{src}^i, x_{src}^i)\}_{i=1}^{N_{batch}}$ and $N_{batch}$ target samples $\{(o_{tgt}^j, x_{tgt}^j)\}_{j=1}^{N_{batch}}$. Let their empirical distributions in the joint $(f_\phi(o), x)$ space be $\hat{\mu}_{src}$ and $\hat{\mu}_{tgt}$. Our UOT loss, $L_{\text{UOT}}(f_\phi)$, is based on the Kantorovich formulation with entropic regularization and KL-divergence penalties for marginal relaxation:

$$L_{\text{UOT}}(f_\phi) = \min_{\Pi \in \mathbb{R}_+^{N_{batch} \times N_{batch}}} \langle \Pi, \hat{C}_\phi \rangle_F + \epsilon \cdot \Omega(\Pi) + \tau \cdot \text{KL}(\Pi \mathbf{1} || \mathbf{p}) + \tau \cdot \text{KL}(\Pi^\top \mathbf{1} || \mathbf{q}). \quad (3)$$

Here, $\hat{C}_\phi$ is the $N_{batch} \times N_{batch}$ ground cost matrix, where each element $(\hat{C}_\phi)_{ij}$ is computed using the joint ground cost described in Sec. 4.1. The term $\Pi$ is the transport plan; $\epsilon > 0$ is the entropic regularization strength with $\Omega(\Pi) = \sum_{i,j} \Pi_{ij} \log \Pi_{ij}$ being the entropy, facilitating efficient solution

via algorithms like Sinkhorn-Knopp [49]; $\tau > 0$ controls the penalty for deviating from the batch marginals $\mathbf{p}$ and $\mathbf{q}$ (typically uniform); and $\mathrm{KL}(\cdot||\cdot)$ denotes the Kullback-Leibler divergence.

### 4.3 Temporally Aligned Sampling for Effective Mini-Batch Learning

The efficacy of mini-batch OT, including UOT, hinges on presenting the solver with comparable source and target samples within each batch. For sequential robotic data, naive random sampling of individual transitions from $D_{src}$ and $D_{tgt}$ may yield pairs from different stages of tasks, leading to noisy transport plans and sub-optimal feature alignment by $f_\phi$. Increasing the minibatch size may lead to higher likelihood of sampling aligned pairs but requires more computational resources.

To address this, we introduce a *temporally aligned sampling* strategy designed to construct mini-batches with a higher density of meaningfully corresponding state-pairs. Our strategy leverages trajectory-level similarity as a heuristic. We first quantify similarity between source trajectories $\{\xi_{src}^k\} \subset D_{src}$ and target trajectories $\{\xi_{tgt}^l\} \subset D_{tgt}$ using Dynamic Time Warping (DTW) [50] on their respective proprioceptive state sequences $\{x_t\}$. The resulting normalized DTW distance, $\bar{d}(\xi_{src}^k, \xi_{tgt}^l) = d_{\mathrm{DTW}}(\xi_{src}^k, \xi_{tgt}^l)/\max(|\xi_{src}^k|, |\xi_{tgt}^l|)$, reflects overall behavioral similarity. To turn these distances into sampling weights, we apply a softplus-based transformation: $w(\xi_{src}^k, \xi_{tgt}^l) = 1/(1 + e^{10 \cdot (\bar{d}(\xi_{src}^k, \xi_{tgt}^l) - 0.01)})$. Mini-batch construction then proceeds by (1) sampling a pair of trajectories $(\xi_{src}, \xi_{tgt})$ with probability biased towards pairs exhibiting high similarity (i.e., low DTW distance) and (2) subsequently sampling individual transition tuples $(o_{src}, x_{src}, a_{src})$ and $(o_{tgt}, x_{tgt}, a_{tgt})$ from this selected, behaviorally similar trajectory pair.

Fine-grained temporal alignment, such as sampling around DTW-matched time steps, can optionally be employed here. We describe how the UOT loss (Eq. 3) is adapted with this new sampling procedure in the appendix.

This two-stage process significantly increases the likelihood that source and target samples within a mini-batch share similar proprioceptive states $x$. Consequently, the UOT optimization (Eq. 3) can more effectively focus on aligning the visual latent features $f_\phi(o_{src})$ and $f_\phi(o_{tgt})$ for these relevant state-pairs. We empirically verify the importance of this sampling strategy in appendix.

### 4.4 Joint Co-Training Framework

Putting all components together, our final approach is a joint co-training framework where the visual feature encoder $f_\phi$ and the policy $\pi_\theta(a|z, x)$ are optimized concurrently. The Unbalanced Optimal Transport loss ($L_{\mathrm{UOT}}$) serves as a regularization term, guiding $f_\phi$ to learn domain-invariant and action-relevant latent representations $z = f_\phi(o)$, while standard Behavior Cloning (BC) losses drive the policy learning. The overall training objective $L(f_\phi, \pi_\theta) = L_{\mathrm{BC}}(f_\phi, \pi_\theta) + \lambda \cdot L_{\mathrm{UOT}}(f_\phi)$ combines these components, where $L_{\mathrm{BC}}$ represents the combined behavior cloning losses calculated over both source ($D_{src}$) and target ($D_{tgt}$) datasets using a standard imitation loss (e.g., MSE). The hyper-parameter $\lambda > 0$ balances feature alignment with policy imitation. The $L_{\mathrm{UOT}}(f_\phi)$ term is computed as defined in Equation 3, with mini-batches sampled with strategy described in Sec. 4.3. The overall training process is detailed in the appendix.

## 5 Experiments

We aim to validate the following core hypotheses. **H1**: Our method effectively learns complex manipulation tasks in both simulation and the real world. **H2**: Our method generalizes to target domains only seen in simulation. **H3**: Our method is broadly applicable to multiple observation modalities. **H4**: Scaling up simulation data coverage improves generalization performance.

### 5.1 Experiment Setups

To evaluate the effectiveness of our approach, we conduct comprehensive experiments in both sim-to-sim and sim-to-real transfer scenarios on a suite of robotic tabletop manipulation tasks: `Lift`, `BoxInBin`, `Stack`, `Square`, `MugHang`, and `Drawer`. These tasks are designed to test the system's ability to handle key challenges in robotic manipulation, including dense object interactions, long-horizon reasoning, and high-precision control.

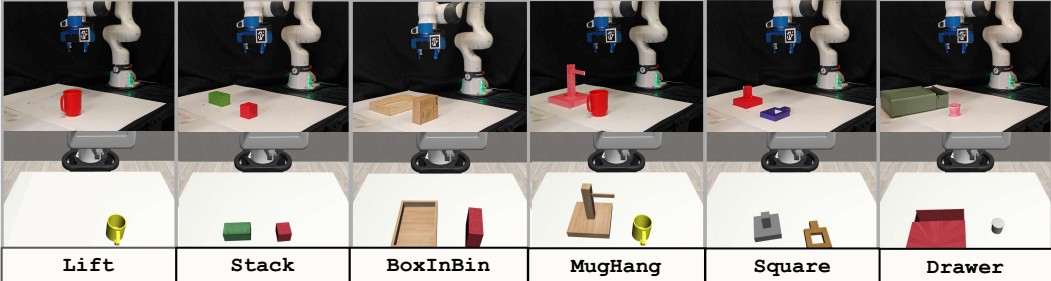

| Lift | Stack | BoxInBin | MugHang | Square | Drawer |

Figure 3: **Evaluation Task Suites.** We evaluate our methods on 6 different tasks in the real world (*top*) and simulation (*bottom*) to demonstrate the effectiveness of sim-to-real transfer.

**Environment setup.** For the real-world experiments, we deploy our method on a Franka Emika Panda robotic arm, controlled at 20Hz using joint impedance control. Visual and point cloud observations are captured using an Intel RealSense D435 depth camera. Our simulation environments use the Robosuite [51] simulation framework. We calibrate the camera pose and camera intrinsics in simulation to match those obtained from the real-world setup to reduce the domain gap.

**Simulation data.** For simulation experiments, we begin by collecting 10 human demonstrations per task. Using MimicGen [10], we synthesize 200-1000 trajectories in the source domain, covering the full range of initial states (denoted as `Source`). In the target domain, we divide the reset region into two subregions: one is populated with 10 trajectories for training (denoted as `Target`), while the other remains completely held out from training (denoted as `Target-OOD`). This held-out subregion is used to evaluate each method's generalization under Out-Of-Distribution (OOD) conditions.

**Real data.** For real-world experiments, we adopt a similar strategy by partitioning the reset region—aligned with the simulation setup into two subregions. Based on task complexity, we collect 10–25 human demonstrations within one subregion and generate 1000 simulated trajectories. To evaluate generalization in OOD scenarios, we consider the following settings in the real world: `Shape`, where the test object has not been seen during real data collection; `Reset`, where the initial object pose falls outside the range covered by demonstrations; and `Texture`, where the object is wrapped in a novel texture not present in any real-world training data. Detailed visualizations of each task and reset configuration are included in the appendix.

**Observation modality and domain gaps.** We evaluate our approach using two observation modalities: point clouds and RGB images. For point cloud observations, our method and the baselines adapt 3D Diffusion Policy [28] with a PointNet encoder [52]. For RGB image observations, we use Diffusion Policy [53] with a ResNet-18 encoder [54]. To evaluate the generalization capabilities of different methods under visual domain shifts in simulation, we introduce several target domain variations: `Viewpoint1-Point`, `Viewpoint3-Point`, `Perturbation-Point`, `Viewpoint-Image`, and `Texture-Image`. Descriptions of each domain shift are provided in the appendix.

**Baselines.** We compare our method against the following baselines: `MMD`—minimizes the distance between the mean embeddings of source and target data [18]; `Co-training`—trains the model using a mixed batch of source and target domain data, following the strategy proposed by [21, 22]; `Source-only`—trains the model exclusively on data from the source domain, which in sim-to-real experiments corresponds to using only simulation data; `Target-only`—trains the model exclusively on data from the target domain, which in sim-to-real experiments trains with only real world data.

## 5.2 Cross-Domain Generalization Results

| | Stack (V) | | Square (V) | | BoxInBin (V) | | Stack (T) | | Square (T) | | BoxInBin (T) | | **Average** | |
| | **T** | **T-O** | **T** | **T-O** | **T** | **T-O** | **T** | **T-O** | **T** | **T-O** | **T** | **T-O** | **T** | **T-O** |
|---|---|---|---|---|---|---|---|---|---|---|---|---|---|---|
| S.-only | 0.00 | 0.00 | 0.00 | 0.00 | 0.00 | 0.00 | 0.00 | 0.00 | 0.00 | 0.00 | 0.00 | 0.00 | 0.00 | 0.00 |
| T.-only | 0.30 | 0.00 | 0.20 | 0.00 | 0.82 | 0.00 | 0.42 | 0.00 | 0.48 | 0.00 | 0.64 | 0.00 | 0.48 | 0.00 |
| MMD | 0.38 | 0.00 | 0.18 | **0.04** | 0.82 | 0.16 | 0.44 | 0.4 | 0.38 | 0.34 | 0.80 | 0.70 | 0.50 | 0.30 |
| Co-train. | 0.44 | **0.04** | 0.76 | 0.00 | **0.90** | 0.14 | 0.54 | 0.34 | 0.66 | 0.46 | **0.98** | 0.72 | 0.71 | 0.28 |
| Ours | **0.65** | **0.04** | **0.86** | 0.02 | 0.88 | **0.26** | **0.66** | **0.52** | **0.68** | **0.54** | 0.96 | **0.82** | **0.78** | **0.36** |

Table 1: **Sim-to-Sim Success Rates for Image-Based Policies**. V and T represent `Viewpoint-Image` and `Texture-Image` domain shifts, respectively. **T** and **T-O** correspond to the target domain and target domain with out-of-distribution (OOD) scenarios.

| | Stack (R) | | Square (R) | | BoxInBin (T) | | **Average** |
| | grasp | full | grasp | full | grasp | full | full |
|---|---|---|---|---|---|---|---|
| Source-only | 0.0 | 0.0 | 0.0 | 0.0 | 0.0 | 0.0 | 0.00 |
| Target-only | 0.0 | 0.0 | 0.1 | 0.0 | 0.0 | 0.0 | 0.00 |
| Co-training | 0.0 | 0.0 | 0.3 | 0.0 | 0.4 | 0.3 | 0.10 |
| Ours | **0.4** | **0.4** | **0.5** | **0.1** | **0.7** | **0.7** | **0.40** |

Table 2: **Real-World Image-Based Policy OOD Success Rates.** R and T denote `Reset` OOD and `Texture` OOD, respectively. The **Average** denotes the average full task success rates over all tasks.

| | BoxInBin (V3) | | BoxInBin (P) | | Lift (V1) | | Stack (V1) | | Square (V1) | | MugHang (V1) | | **Average** | |
| | T | T-O | T | T-O | T | T-O | T | T-O | T | T-O | T | T-O | T | T-O |
|---|---|---|---|---|---|---|---|---|---|---|---|---|---|---|
| S.-only | 0.08 | 0.10 | 0.52 | 0.60 | 0.32 | 0.40 | 0.52 | 0.64 | 0.10 | 0.08 | 0.12 | 0.10 | 0.28 | 0.32 |
| T.-only | 0.42 | 0.00 | 0.58 | 0.00 | 0.60 | 0.00 | 0.32 | 0.00 | 0.16 | 0.00 | 0.18 | 0.00 | 0.38 | 0.00 |
| MMD | 0.50 | 0.38 | 0.66 | 0.50 | 0.56 | 0.52 | 0.70 | 0.66 | 0.18 | 0.12 | 0.18 | 0.20 | 0.46 | 0.40 |
| Co-train. | 0.76 | 0.52 | 0.70 | 0.66 | **0.92** | 0.48 | **0.86** | 0.72 | 0.24 | 0.24 | 0.26 | 0.22 | 0.62 | 0.47 |
| Ours | **0.84** | **0.58** | **0.80** | **0.76** | 0.80 | **0.60** | 0.82 | **0.86** | **0.42** | **0.38** | **0.40** | **0.34** | **0.68** | **0.59** |

Table 3: **Sim-to-sim Success Rates For Point Cloud-Based Policies.** V1, V3, and P indicate domain shifts due to `Viewpoint1-Point`, `Viewpoint3-Point`, and `Perturbation-Point`, respectively. **T** and **T-O** denote the target domain and target domain under out-of-distribution (OOD) conditions.

We report results for policies using point cloud and image-based observations in both simulation and real-world settings. For simulation, image based and point cloud based performance are shown in Tables 1 and 3. For real-world experiments, in-distribution results are presented in Tables 7 and 8 in Appendix, while out-of-distribution (OOD) performance is reported in Tables 2 and 4.

**These results support the following key hypotheses:**

**Our method effectively learns complex manipulation tasks in both simulation and the real world (H1).** Experimental results show that our approach consistently matches or outperforms in terms of success rates all baselines across source and target domains in both simulated and real-world settings. On real-world tasks, our method achieves average success rates of 0.73 and 0.77 for image-based and point cloud-based policies, respectively. The `Target-only` baseline performs well in distribution but fails to generalize under domain shifts. The `MMD` baseline [18] offers limited improvement by aligning global feature statistics, but its coarse alignment often disrupts task-relevant structure and harms source-domain performance. In contrast, by using Unbalanced Optimal Transport, our method performs selective, structure-aware alignment, avoiding spurious matches.

**Our method generalizes to target domains only seen in simulation (H2).** In `Target-OOD` scenarios, our method outperforms all baselines, underscoring the value of learning domain-invariant representations (see Tab. 2 and Tab. 4). While `Co-training` baselines [21, 22] perform well when the target-domain training data overlaps with the evaluation region, they struggle to generalize when this overlap is absent. This limitation is especially evident under large domain shifts. For example, in the real-world `BoxInBin` and `Stack` tasks with novel textures or reset poses, our method achieves success rates of 0.7 and 0.4, respectively, using image-based observations—compared to just 0.3 and 0.0 for the `Co-training` baseline. These results highlight the shortcomings of relying purely on supervised target-domain data without explicitly addressing domain shift.

**Our method is broadly applicable to multiple observation modalities (H3).** We observe consistent performance gains across both image-based and point cloud inputs. In simulation, policies trained with either modality outperform all baselines, demonstrating that our approach effectively learns domain-invariant features that capture task-relevant information on multiple sensory modalities.

**Simulation data provides a scalable and effective way to augment real-world training.** Across real-world tasks, both our method and the `Co-training` baseline benefit significantly from augmenting limited real-world demonstrations with simulated data. Policies trained with this augmented data consistently outperform the `Target-only` baseline, especially in out-of-distribution (OOD) settings where real-world coverage is sparse. This underscores the value of using low-cost simulation data to fill in gaps in real-world datasets, enabling more scalable and generalizable behavior cloning.

**Scaling up simulation data coverage improves real-world performance (H4).** To analyze simulation data scaling, we consider the `Stack` task in the real world with point cloud observation. We generated 100, 300, 500, and 1000 simulated trajectories, combined them with 25 real-world

| | Stack (R) | | Square (R) | | BoxInBin (R) | | Lift (R) | | Lift (S) | | Lift (R+S) | | **Average** |
| --- | --- | --- | --- | --- | --- | --- | --- | --- | --- | --- | --- | --- | --- |
| | grasp | full | grasp | full | grasp | full | reach | full | reach | full | reach | full | full |
| S.-only | 0.6 | 0.3 | 0.1 | 0.1 | 0.3 | 0.2 | 0.8 | 0.8 | 0.6 | 0.6 | 0.9 | 0.9 | 0.48 |
| T.-only | 0.0 | 0.0 | 0.3 | 0.0 | 0.0 | 0.0 | 0.0 | 0.0 | **1.0** | **1.0** | 0.0 | 0.0 | 0.17 |
| Co-train. | 0.4 | 0.1 | **0.6** | **0.2** | 0.2 | 0.0 | 0.8 | 0.8 | **1.0** | **1.0** | 0.9 | 0.9 | 0.50 |
| Ours | **0.8** | **0.4** | **0.6** | 0.1 | **0.7** | **0.5** | **1.0** | **1.0** | **1.0** | **1.0** | **1.0** | **1.0** | **0.67** |

Table 4: **Real World Point-Cloud-Based Policy OOD Success Rates.** R and S denote Reset OOD and Shape OOD, respectively. The **Average** denotes the average full task success rates over all tasks.

demonstrations, and trained both our method and the Co-training baseline. As shown in Fig. 4(b), increasing the amount of simulation data significantly improves our method's performance in target domain regions that lack real-world coverage, highlighting the importance of learning a domain-invariant latent space that enables the policy to generalize beyond observed distributions.

**Our method learns shared latent space that aligns simulation and real data.** To better understand how the learned embeddings contribute to generalization, we visualize them using t-SNE [55], as shown in Fig. 4(a). Blue and red correspond to features extracted from source and target domain observations, respectively. The left plot shows embeddings from the encoder trained with the Co-training baseline, while the right plot shows embeddings from our method. The visualization reveals that our approach leads to significantly better alignment between source and target distributions, highlighting its ability to learn domain-invariant representations that facilitate robust generalization. We also visualize the transport plan on a randomly sampled batch, along with the corresponding image observations from the source and target domains. As shown in Fig. 10 in the Appendix, the transport plan effectively aligns data points with similar states across domains.

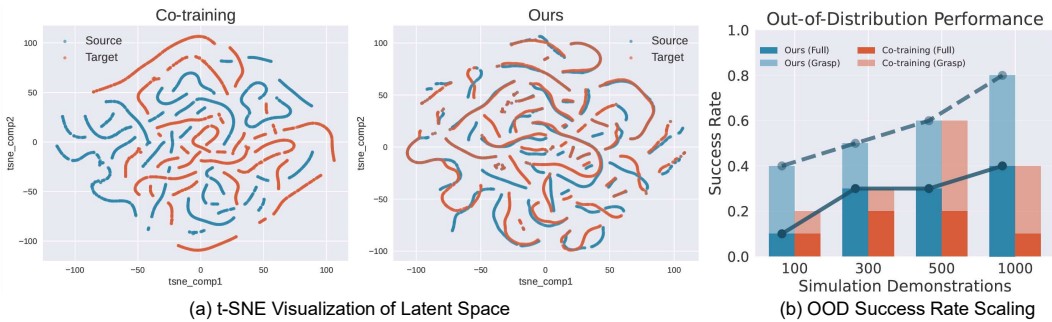

(a) t-SNE Visualization of Latent Space    (b) OOD Success Rate Scaling

Figure 4: (a) **Latent Space Visualization.** Our OT alignment maps source domain samples (blue) and target domain samples (red) nearby in the latent space, yielding a single, well-mixed cluster. This overlap demonstrates that OT alignment effectively synchronizes cross-domain feature distributions, improving sim-to-real transfer. (b) **Out-Of-Distribution Performance.** Scaling the number of simulation demonstrations leads to significant OOD success rate gains.

## 6 Limitations and Conclusions

The main limitation of our work is that we only address sim-to-real visual observation gaps. Addressing action dynamics gaps remains future work. Because we use MimicGen [10] for automated simulation demonstration generation, we inherit its limitations, namely, our method primarily applies to tasks with prehensile and quasi-static interactions. We require a small number of real-world on-task demonstrations that are aligned with the simulated demonstrations. Future work involves relaxing this requirement, for example, by instead consuming unstructured real-world data, such as play data.

In conclusion, we presented a framework for effectively incorporating large datasets of simulation demonstrations into real-world policy learning pipelines via feature-consistent co-training. We proposed using Optimal Transport (OT) to align encoder features to be invariant to whether observations are from simulation or the real world, improving the transferability of simulation data. Because we have much more simulation data than real-world data, we incorporated an Unbalanced OT loss within our training objective and devised a data sampling scheme that explicitly yields similar simulation and real-world demonstration pairs. Finally, we demonstrated the improved learning performance arising from the simulated demonstrations both in a sim-to-sim testbed as well as in real-world tasks.

# 7   Acknowledgment

This work was supported in part by the National Science Foundation under Awards No. 2409016 and 2442393. Any opinions, findings, and conclusions or recommendations expressed in this material are those of the author(s) and do not necessarily reflect the views of the National Science Foundation.

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

# A    Table of Contents

The supplementary material has the following contents:

- **Task and Hardware Setups** (Sec. B): Detailed descriptions of the hardware setups, dataset, and task settings.
- **Model and Training Details** (Sec. C): Descriptions of the neural network architectures used in our experiments and the corresponding training procedures.
- **Ablation Study on Sampling Strategy** (Sec. D): Evaluation and analysis of different sampling strategies.
- **Additional Results** (Sec. F): In-distribution evaluation results for both image-based and point cloud-based policies in the real world.
- **Transport Plan Visualization** (Sec. H): Visualizations of the optimal transport plan for randomly sampled training batches.
- **Visualization of Latent Space** (Sec. I): Visual comparisons of the learned latent spaces between our method and the `Co-training` baseline across additional tasks.

More video results and analysis can be found on our website: `https://ot-sim2real.github.io/`

# B    Task and Hardware Setups

To evaluate the effectiveness of our approach, we conduct comprehensive experiments on a suite of robotic tabletop manipulation tasks, covering both sim-to-sim and sim-to-real transfer scenarios. These tasks are designed to test the system's ability to handle key challenges in robotic manipulation, including dense object interactions, long-horizon reasoning, and high-precision control:

- `Lift`: Grasp the rim of a mug and lift it vertically;
- `BoxInBin`: Grasp a tall box and place it into a bin;
- `Stack`: Grasp a small cube and stack it on top of a longer cuboid;
- `Square`: Grasp the handle of a square-shaped object and insert it onto a peg;
- `MugHang`: Grasp the rim of a mug and hang it on a mug tree using the handle;
- `Drawer`: Open a drawer, grasp a coffee pod from the table, place it into the drawer, and close the drawer.

## B.1    Hardware Setups

The system setup is illustrated in Fig. 5. We use a Franka Emika Panda robot controlled via a joint impedance controller [56] running at 20 Hz for policy execution. For data collection, the robot is teleoperated using a Meta Quest 3 headset, with tracked Cartesian poses converted to joint configurations through inverse kinematics. RGB image and depth are captured using an Intel RealSense D435 depth camera.

## B.2    Domain Shifts and Observation Gaps

We assess generalization under visual domain shifts in simulation through designing the following target domain shifts:

- `Viewpoint1-Point`: The camera is rotated approximately $30°$ around the z-axis, resulting in a side view in the target domain compared to a front-facing view in the source. Point cloud observations are used.
- `Viewpoint3-Point`: The camera is rotated approximately $90°$ around the z-axis, introducing a more extreme viewpoint shift. Point cloud observations are used.
- `Perturbation-Point`: Random noise sampled uniformly from the range $[-0.01, 0.01]$ is added to each point in the point cloud to simulate sensor noise or domain shift.

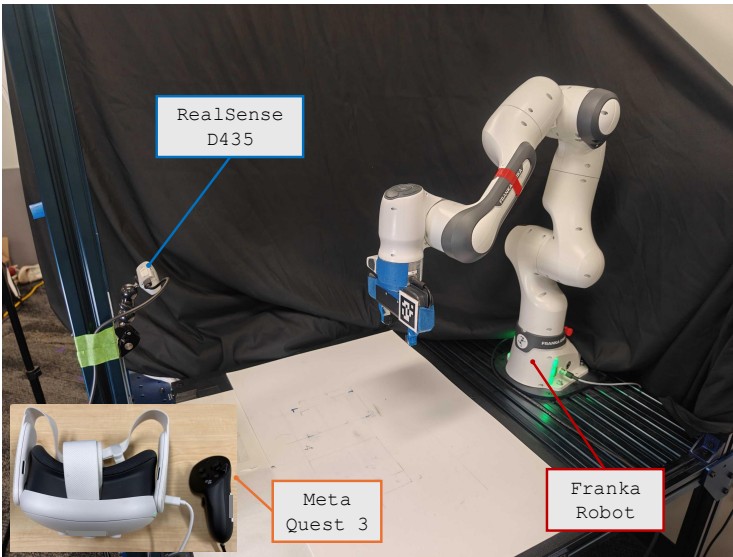

Figure 5: **Hardware Setup.** Our hardware platform uses a Franka Emika Panda robot, with an Intel RealSense D435 camera for capturing image and depth, and a Meta Quest 3 headset for teleoperation.

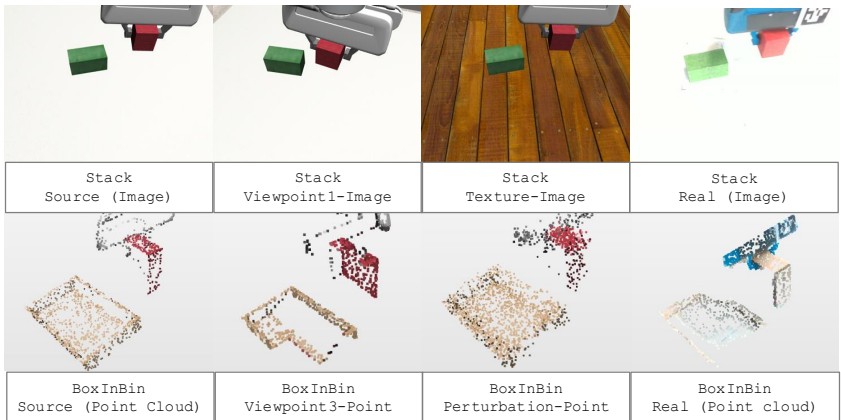

Figure 6: **Observation Gap Across Domains.** Top: image observations for the `Stack` task from source, `Viewpoint1-Image`, `Texture-Image`, and real-world domains. Bottom: point cloud observations for the `BoxInBin` task from source, `Viewpoint3-Point`, `Perturbation-Point`, and real-world domains. Point cloud color is for visualization only and not used as input to the policy.

- `Viewpoint1-Image`: A $20°$ camera rotation around the z-axis is applied. RGB image observations are used.
- `Texture-Image`: The table texture in the target domain is modified. RGB image observations are used.

We illustrate the observation gap across all domains in Fig. 6. The first row displays image observations for the `Stack` task from the source domain, `Viewpoint1-Image`, `Texture-Image`, and the real world. The second row shows point cloud observations for the `BoxInBin` task from the source domain, `Viewpoint3-Point`, `Perturbation-Point`, and the real world. Point cloud color is for visualization only and not used as input to the policy.

### B.3 Task Datasets, Reset Ranges, and OOD Variants

We focus primarily on evaluating policy performance in regions covered exclusively by source-domain demonstrations. To conduct controlled experiments, we define three distinct reset regions for each task—`Source`, `Target`, and `Target-OOD`—as shown in Fig. 7. Specifically:

| | Lift | Stack | BoxInBin | MugHang | Square | Drawer |
|---|---|---|---|---|---|---|
| Number of real demos | 10 | 25 | 20 | 15 | 25 | 25 |

Table 5: **Number of Real-World Demonstrations.** We collect 10–25 demonstrations per task, varying with task difficulty.

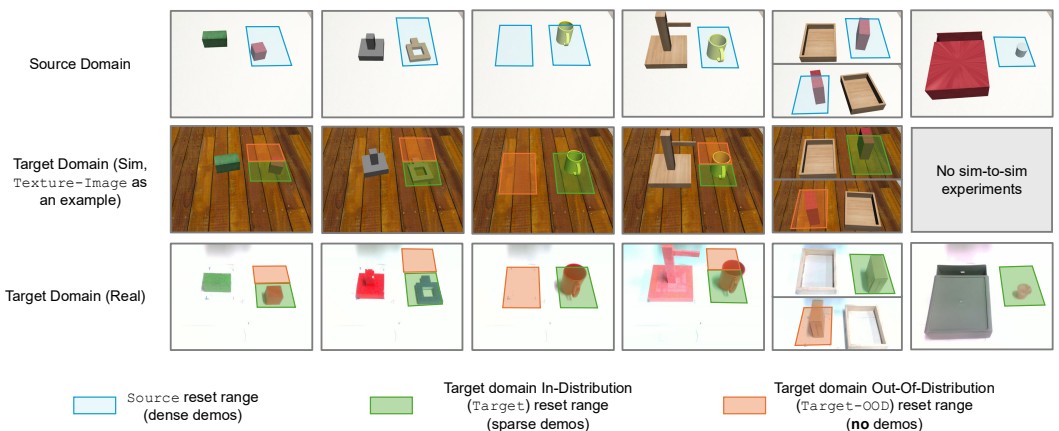

Figure 7: **Reset Ranges for Each Task.** The first row illustrates the Source region, where dense source-domain demonstrations are collected. The second row shows the Target and Target-OOD reset ranges used in sim-to-sim transfer experiments. In this setting, the Target region is sparsely covered by demonstrations, while the Target-OOD region contains no demonstrations and is used exclusively for policy evaluation. The third row similarly presents the Target and Target-OOD regions for sim-to-real transfer experiments.

- Source: A large region that is densely covered by demonstrations in the source domain. We generate 1000 demonstrations using MimicGen [10] within the Source region.

- Target: A small subset of the Source region. This region is sparsely covered by demonstrations in the target domain, and is therefore considered in-distribution during evaluation. For sim-to-sim transfer, we collect 10 demonstrations within this region. For sim-to-real transfer, the number of real-world demonstrations collected in the Target region is adjusted based on task difficulty, as detailed in Tab. 5.

- Target-OOD: No demonstrations are collected in the Target-OOD region, which is used solely for evaluation and treated as out-of-distribution (OOD).

For sim-to-real transfer experiments, in addition to the reset range OOD (denoted as Reset), we consider two additional OOD variants. In the Texture variant, the object's texture is modified to one that is unseen in the real-world demonstrations. In the Shape variant, the object is replaced with a novel shape not encountered in the real-world demonstrations. These variants are illustrated in Fig. 8.

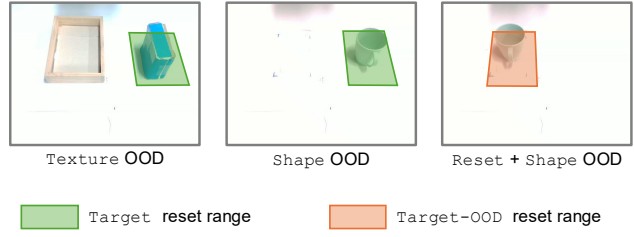

Figure 8: **Texture and Shape OOD in Sim-to-Real Experiments.** Visualization of reset ranges for the BoxInBin task under Texture OOD, and the Lift task under both Shape OOD and Shape+Reset OOD conditions.

## C  Model and Training Details

For point cloud-based experiments, we adopt the 3D Diffusion Policy architecture [28] with a PointNet encoder [52]. The diffusion head receives features extracted from the point cloud observations along with robot proprioceptive inputs (joint and gripper positions), and outputs 7-DOF target joint positions and the gripper action. We project the depth map into the robot base frame to generate the scene point cloud. For a pixel with coordinate $(u, v)$ and depth $d$, the corresponding 3D location can be recovered by:

$$p^w = R \cdot K^{-1} \cdot I + t$$

where $I = (u \cdot d, v \cdot d, d)$, $[R \mid t]$ denotes the camera pose obtained through hand-eye calibration [57], and $K$ denotes the camera intrinsic matrix. We crop the reconstructed scene point cloud using a bounding box defined by $x \in [-0.2, 0.1]$, $y \in [-0.2, 0.2]$, and $z \in [0.008, 0.588]$ to exclude irrelevant background information. The cropped point cloud is then downsampled to 2048 points using Farthest Point Sampling (FPS) [58].

For experiments with image-based policy, we adopt Diffusion Policy [27] with a ResNet-based [54] visual encoder. The original images are captured by the camera at a resolution of $480 \times 640$. During preprocessing, the images are downsampled to $120 \times 160$, followed by random cropping to $108 \times 144$ during training and center cropping during testing. The policy takes stacked history images and robot proprioceptive inputs (joint and gripper positions) as input, and outputs 7-DOF target joint positions along with the gripper action.

Our overall training procedure is summarized in Algm. 1. We use a batch size of 256 for the behavior cloning loss $L_{\text{BC}}$, with a co-training ratio of 0.9 following Maddukuri et al. [22]. For the optimal transport loss $L_{\text{OT}}$, the batch size is set to 128, with a weighting coefficient $\lambda = 0.1$. We use $\epsilon = 0.0005$ and $\tau = 0.01$ in our experiments.

---

**Algorithm 1** Joint Policy Training with OT

---

**Require:** Source dataset $D_{src}$, Target dataset $D_{tgt}$
 1: Initialize encoder $f_\phi$, and policy $\pi_\theta$
 2: Compute DTW distances for all trajectories pairs in $D_{src}$ and $D_{tgt}$
 3: **for** iteration $t = 1$ to $T$ **do**
 4:     Sample a paired batch $\{(o^i_{src}, x^i_{src}, a^i_{src}, o^j_{tgt}, x^j_{tgt}, a^j_{tgt})\}$ with size $N$ from $D_{src}$ and $D_{tgt}$ using strategy described in Sec. 4.3
 5:     Compute features $\{z^i_{src}\}$ and $\{z^j_{tgt}\}$ using encoder $f_\phi$
 6:     Construct ground cost matrix $\hat{C}_\phi$ as described in Sec. 4.1
 7:     Compute optimal transport plan $\Pi^* = \arg\min_{\Pi \in \mathbb{R}^{N \times N}_+}(\langle \Pi, \hat{C}_\phi \rangle_F + \epsilon \cdot \Omega(\Pi) + \tau \cdot \text{KL}(\Pi\mathbf{1}||\mathbf{p}) + \tau \cdot \text{KL}(\Pi^\top\mathbf{1}||\mathbf{q}))$ via Sinkhorn-Knopp algorithm [49]
 8:     Compute OT loss $L_{\text{UOT}}(f_\phi) = \langle \Pi^*, \hat{C}_\phi \rangle_F$
 9:     Sample $\{(o^i_{src}, x^i_{src}, a^i_{src})\}$ from $D_{src}$ and sample $\{(o^j_{tgt}, x^j_{tgt}, a^j_{tgt})\}$ from $D_{tgt}$
10:     Compute BC loss $L_{\text{BC}}(f_\phi, \pi_\theta)$
11:     Update $f_\phi$ and $\pi_\theta$ with gradients of $L_{\text{BC}}(f_\phi, \pi_\theta) + \lambda \cdot L_{\text{UOT}}(f_\phi)$
12: **end for**

---

## D  Ablation Study on Sampling Strategy

To assess the effectiveness of our sampling strategy, we compare the full method against a variant (denoted as `Ours w/o Sampler`) that does not apply any trajectory-level sampling. In this baseline, source and target data are randomly sampled across trajectories and time steps, with no coordination. We also include an oracle variant (denoted as `UOT-Oracle`), which constructs perfectly paired batches—each state is observed in both the source and target domains to ensure that batch data originates from the same underlying states. We evaluate policy performance on the `Stack` task under the `Viewpoint1-Point` variation, with results shown in Fig. 9.

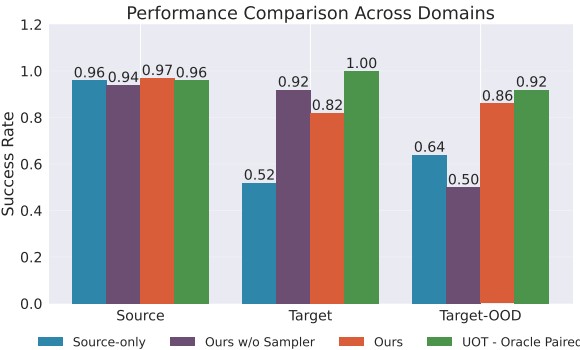

Figure 9: **Sampling Strategies Comparison.** Our proposed sampling strategy (`Ours`) improves policy success rates on the `Stack` task with `Viewpoint1-Point`, outperforming `Ours w/o Sampler`, and achieving performance comparable to the oracle-paired upper bound (`UOT-Oracle`).

**Temporal-aware strategy improves pairing quality and downstream performance.** The oracle baseline demonstrates that, given perfectly aligned data, unbalanced OT loss significantly enhances generalization by enabling the encoder to learn domain-invariant representations. In contrast, the no-sampling variant (`Ours w/o Sampler`) exhibits poor generalization in the `Target-OOD` setting. This degradation likely stems from the low probability of encountering aligned state pairs in mini-batches—especially problematic in long-horizon tasks, where uncoordinated sampling rarely produces temporally aligned data.

# E   Hyperparameter Sensitivity Analysis

We conduct an ablation study to evaluate the sensitivity of our method to key hyperparameters, including the entropy regularization coefficient ($\epsilon$), the KL divergence penalty term ($\tau$), and the window size used in temporally aligned sampling. In each experiment, we vary a single hyperparameter while keeping the others fixed, train the policy, and assess its performance via rollouts. We report results for the `BoxInBin` task under the `Viewpoint-Image` setting and the `Lift` task under the `Texture-Image` setting, as shown in Table 6. The results indicate that our method is robust to hyperparameter variations within reasonable ranges. Specifically, performance remains stable when $\epsilon$ and $\tau$ are set between 0.001 and 0.1, and when the window size is varied between 5 and 20. Our method consistently outperforms the co-training baseline in OOD scenarios, where the baseline achieves a success rate of 0.14 on `BoxInBin` and 0.6 on `Lift`. These findings suggest that, although our approach introduces additional components, it does not require extensive tuning and offers clear advantages in terms of generalization.

# F   Additional Real-world Evaluation Results

We conduct extensive real-world evaluations to validate the effectiveness of our approach. Sim-to-real transfer results for in-distribution scenarios are reported in Tab.7 and Tab.8 for image-based and point cloud-based policies, respectively. Results for out-of-distribution (OOD) scenarios are presented in the main paper.

Experimental results show that our approach consistently outperforms all baselines in real-world in-distribution settings. Our method achieves average success rates of 0.73 and 0.77 for image-based and point cloud-based policies, respectively, demonstrating its effectiveness in learning complex real-world manipulation tasks.

# G   Performance with Scarce Target Domain Data

To assess the effectiveness of our method under limited target domain data, we compare it against several baselines in the low-data regime for the `BoxInBin` task with the `Viewpoint3-Point` setting.

**(a) BoxInBin**

| | $\epsilon$ | | | | | |
|---|---|---|---|---|---|---|
| | 0.0001 | 0.001 | 0.005 | 0.01 | 0.04 | 1 |
| T | 0.90 | 0.94 | 0.92 | 0.88 | 0.90 | 0.88 |
| T-O | 0.18 | 0.16 | 0.26 | 0.22 | 0.18 | 0.20 |

| | $\tau$ | | | | | |
|---|---|---|---|---|---|---|
| | 0.0001 | 0.001 | 0.005 | 0.02 | 0.04 | 1 |
| T | 0.88 | 0.96 | 0.94 | 0.94 | 0.92 | 0.94 |
| T-O | 0.28 | 0.26 | 0.20 | 0.28 | 0.22 | 0.22 |

| | winsize | | | | | |
|---|---|---|---|---|---|---|
| | 1 | 5 | 10 | 20 | 40 | 120 |
| T | 0.82 | 0.92 | 0.86 | 0.90 | 0.94 | 0.84 |
| T-O | 0.20 | 0.22 | 0.22 | 0.24 | 0.16 | 0.14 |

**(b) Lift**

| | $\epsilon$ | | | | | |
|---|---|---|---|---|---|---|
| | 0.0001 | 0.001 | 0.01 | 0.04 | 0.1 | 1 |
| T | 0.84 | 0.88 | 0.80 | 0.76 | 0.78 | 0.76 |
| T-O | 0.60 | 0.74 | 0.62 | 0.66 | 0.68 | 0.54 |

| | $\tau$ | | | | | |
|---|---|---|---|---|---|---|
| | 0.0001 | 0.005 | 0.02 | 0.04 | 0.1 | 1 |
| T | 0.78 | 0.70 | 0.76 | 0.76 | 0.78 | 0.74 |
| T-O | 0.56 | 0.67 | 0.64 | 0.66 | 0.62 | 0.66 |

| | winsize | | | | | |
|---|---|---|---|---|---|---|
| | 1 | 5 | 10 | 20 | 40 | 120 |
| T | 0.86 | 0.80 | 0.70 | 0.78 | 0.74 | 0.82 |
| T-O | 0.66 | 0.60 | 0.67 | 0.58 | 0.56 | 0.60 |

Table 6: **Hyperparameter Sensitivity**. In each series of experiments, we vary a single hyperparameter while keeping the others fixed, train the policy, and assess its success rates via rollouts. T and T-O denote the target domain and the target domain under OOD conditions.

| | Stack | | Square | | BoxInBin | | **Average** |
|---|---|---|---|---|---|---|---|
| | grasp | full | grasp | full | grasp | full | full |
| Source-only | 0.1 | 0.0 | 0.0 | 0.0 | 0.0 | 0.0 | 0.00 |
| Target-only | 0.7 | 0.7 | 0.8 | 0.0 | 0.7 | 0.7 | 0.47 |
| Co-training | 0.8 | 0.7 | 0.8 | 0.1 | **0.9** | 0.8 | 0.53 |
| Ours | **0.9** | **0.9** | **0.9** | **0.4** | **0.9** | **0.9** | **0.73** |

Table 7: **Real World Image-Based Policy In-Distribution Success Rates.** The **Average** denotes the average full task success rates over all tasks.

| | Stack | | Square | | BoxInBin | | Lift | | MugHang | | Drawer | | | | **Average** |
|---|---|---|---|---|---|---|---|---|---|---|---|---|---|---|---|
| | grasp | full | grasp | full | grasp | full | reach | full | grasp | full | open | grasp | place | full | full |
| S.-only | 0.3 | 0.0 | 0.1 | 0.1 | 0.4 | 0.3 | 0.5 | 0.5 | 0.1 | 0.0 | 0.0 | 0.0 | 0.0 | 0.0 | 0.15 |
| T.-only | 0.7 | 0.4 | 0.6 | 0.1 | **0.9** | 0.8 | **1.0** | **1.0** | 0.8 | **0.8** | 0.9 | 0.5 | 0.5 | 0.5 | 0.60 |
| Co-train. | 0.7 | 0.7 | **1.0** | **0.5** | 0.8 | 0.8 | 0.8 | 0.8 | **1.0** | **0.8** | **1.0** | 0.7 | 0.7 | 0.4 | 0.67 |
| Ours | **0.8** | **0.8** | **1.0** | 0.4 | **0.9** | **0.9** | **1.0** | **1.0** | **1.0** | **0.8** | **1.0** | 0.7 | 0.7 | **0.7** | **0.77** |

Table 8: **Real World Point-Cloud-Based Policy In-Distribution Success Rates.** The **Average** denotes the average full task success rates over all tasks.

| | 1 Demo | | 5 Demo | |
|---|---|---|---|---|
| | Target | Target-OOD | Target | Target-OOD |
| Ours | **0.56** | **0.28** | **0.70** | **0.32** |
| Co-training | 0.46 | 0.00 | 0.38 | 0.22 |
| MMD | 0.42 | 0.16 | 0.34 | 0.22 |
| Target-only | 0.00 | 0.00 | 0.46 | 0.00 |

Table 9: **Performance with Limited Target Domain Data.** We report success rates for various methods on the BoxInBin task with the Viewpoint3-Point setting, under scenarios where data from the target domain is extremely limited.

As shown in Tab. 9, our approach consistently outperforms the baselines, highlighting its robustness even with extremely limited supervision.

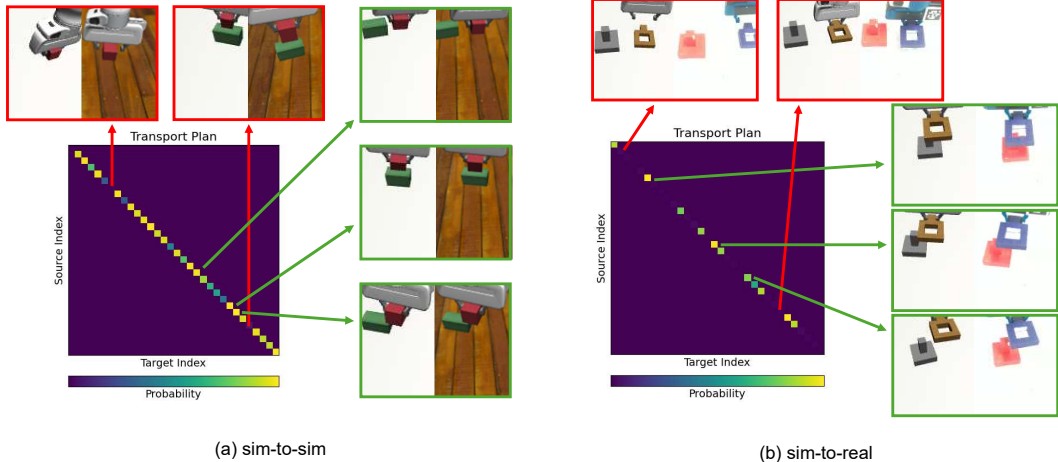

(a) sim-to-sim

(b) sim-to-real

Figure 10: **Transport Plan Visualization.** We visualize the transport plan for a randomly sampled batch during training the image-based policy, alongside corresponding observations from both domains. The left figure shows a sim-to-sim experiment, while the right shows a sim-to-real experiment. The visualization reveals that the transport plan effectively aligns similar states across domains, as indicated by high transport probabilities.

## H    Transport Plan Visualization

To understand how optimal transport facilitates domain-invariant feature learning and enhances cross-domain generalization, we visualize the transport plan for a randomly sampled batch during training the image-based policy, along with corresponding observations from both domains (see Fig. 10). The left plot shows results from the sim-to-sim transfer experiment, while the right plot depicts the sim-to-real setting. The results show that the transport plan effectively aligns similar states across domains, encouraging domain-invariant representations.

## I    Visualization of Latent Space

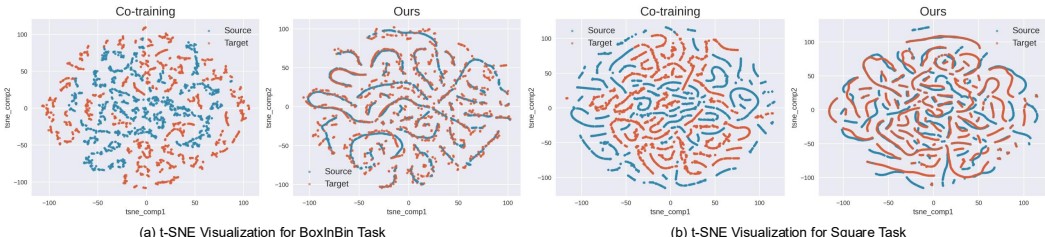

(a) t-SNE Visualization for BoxInBin Task

(b) t-SNE Visualization for Square Task

Figure 11: **Latent Space Visualization.** Latent space comparison between the `Co-training` baseline and our method. In our approach, source-domain points (blue) and target-domain points (red) form a well-mixed cluster, illustrating how OT alignment harmonizes cross-domain feature distributions and enhances transferability and generalization.

Beyond the feature visualization for the `Stack` task with the `Viewpoint1-Point` target domain, we also present additional t-SNE [55] visualizations in Fig. 11 for the `BoxInBin` task with the `Perturbation-Point` target domain, and the `Square` task with the `Viewpoint1-Point` target domain. We compare the latent spaces produced by the `Co-training` baseline and our method. In our approach, source-domain points (blue) and target-domain points (red) form a well-mixed cluster, highlighting how OT alignment harmonizes cross-domain feature distributions and improves both transferability and generalization.

