# OpenReview forum: "Generalizable Domain Adaptation for Sim-and-Real Policy Co-Training"
_NeurIPS.cc/2025/Conference — NeurIPS 2025 poster_

### Official Review · Reviewer_PAX9 · 2025-06-16

**Clarity:** 3
**Significance:** 2
**Originality:** 2
**Rating:** 4
**Confidence:** 4

**Summary:**

The paper proposes a unified co-training framework that trains a single policy on both simulation and real-world demonstrations. The core innovation is the use of an Unbalanced Optimal Transport (UOT) loss to align joint distributions of observations and actions (or proprioceptive states) across domains in a shared latent space, rather than aligning marginal distributions. This action-aware alignment, augmented with temporally aligned sampling, preserves task-relevant features and handles domain imbalance.

The authors evaluate across six manipulation tasks using both RGB and point cloud modalities. They compare against Source-only, Target-only, MMD-based, and naive co-training baselines. Metrics include task success rates in standard and out-of-distribution (OOD) scenarios in both simulated and real environments. Key tools include MimicGen for simulation data and Robosuite for simulation environments.

**Questions:**

1. How sensitive is the performance to the choice of UOT hyperparameters (e.g., entropy regularization, KL penalties)?
2. Could the approach generalize to tasks with significant dynamics gaps, such as high-speed or contact-rich manipulation? It seems all evaluated tasks are transferred kinematically.
3. How does the performance scale with even fewer real-world demonstrations (e.g., <5 per task)?
4. Would alternative domain alignment methods (e.g., contrastive learning or adversarial domain alignment [2, 3]) offer competitive or complementary benefits?

[2] Raychaudhuri, Dripta S., et al. "Cross-domain imitation from observations." International conference on machine learning. PMLR, 2021.

[3] Kim, Kuno, et al. "Domain adaptive imitation learning." International Conference on Machine Learning. PMLR, 2020.

**Ethical Concerns:**

["NO or VERY MINOR ethics concerns only"]

**Final Justification:**

I have read the long rebuttal of the authors, which satisfactorily addressed my questions and concerns on sensitivity analysis and further scalability issues. Hence, I am leaning towards accepting this paper.

**Limitations:**

Yes

**Quality:**

3

**Strengths And Weaknesses:**

**Strengths**
- **Use of UOT:** The integration of UOT into a co-training framework for sim-to-real transfer is methodologically sound and innovative. OT provides a structured and geometry-aware way to align feature-action distributions, which is superior to marginal feature alignment techniques like MMD. The author should cite [1] since it is very relevant on latent-space alignment with UOT without the marginal constraints.
- **Data Imbalance Handling:** UOT’s ability to handle domain imbalance is crucial in practical sim-to-real settings where simulated data vastly outnumbers real-world examples. This mitigates overfitting to the dominant domain and avoids spurious alignments.
- **Improved OOD Generalization:** The method shows consistent performance gains in out-of-distribution (OOD) settings, highlighting its strength in extrapolating from simulated experiences to unseen real-world scenarios. Real-world experiments with the Franka Panda robot validate that the method is not only simulation-effective but deployment-ready for physical systems.

**Weaknesses**

- **Limited Addressing of Dynamics Gap:** The method only tackles the visual observation gap and assumes quasi-static, prehensile tasks. It does not address the sim-to-real dynamics mismatch (e.g., surface friction of push block), which can be substantial in more dynamic or contact-rich scenarios.
- **Dependence on Demonstration Alignment:** The method assumes the overhead of behavioral similarity between simulated and real demonstrations, enforced via DTW. This may not hold in scenarios where real demonstrations deviate significantly in strategy or execution from simulated ones. Moreover, the approach relies on action or proprioceptive similarity across domains. This limits its application to settings where such consistency is hard to guarantee (e.g., different control schemes or hardware platforms).
- **Limited Theoretical Development:** The paper does not present theoretical guarantees or convergence analysis for the proposed UOT-guided co-training. While empirical results are strong, formal properties are not fully explored.
- **Hyperparameter Sensitivity Not Explored:** The paper does not provide an ablation or sensitivity analysis on key UOT parameters (e.g., entropy regularization, KL divergence penalties), which may influence performance stability.

[1] Nguyen, Duy Minh Ho, et al. "Dude: Dual Distribution-Aware Context Prompt Learning For Large Vision-Language Model." Asian Conference on Machine Learning. PMLR, 2025.

---

> ### Author Rebuttal · Authors · 2025-07-31
>
> Thank you for your time and thoughtful feedback. We appreciate the positive recognition of our integrated co-training framework for sim-to-real transfer, which is both methodologically sound and innovative. We are also grateful for the acknowledgment of its ability to handle imbalanced data distributions across domains—a critical challenge in practical real-world applications—and its consistent performance gains in challenging evaluation scenarios, including real-world experiments, a setting often underexplored in prior work. Below, we address the raised concerns in detail:
>
> - **Sensitivity to hyper-parameters:** We conducted an ablation study to evaluate the sensitivity of our method to key hyperparameters, including the entropy regularization coefficient ($\epsilon$), the KL divergence penalty term ($\tau$), and the window size used in temporally aligned sampling. In each experiment, we varied a single hyperparameter while keeping the others fixed, trained the policy, and assessed its performance via rollout evaluations. We report results for the *BoxInBin* task under the *Viewpoint-Image* setting and the *Lift* task under the *Texture-Image* setting, as shown in the following tables (Table R1-R6). The results indicate that **our method is robust to hyperparameter variations** within reasonable ranges. Specifically, performance remains stable when $\epsilon$ and $\tau$ are set between 0.001 and 0.1, and when the window size is varied between 5 and 20. Our method consistently outperforms the co-training baseline in out-of-distribution (OOD) scenarios, where the baseline achieves a success rate of 0.14 on *BoxInBin* and 0.6 on *Lift*. These findings suggest that, although our approach introduces additional components, it does not require extensive tuning and offers clear advantages in terms of generalization.
>
>  $\epsilon$ | 0.0001 | 0.001 | 0.01 | 0.04 |  0.1 |   1  |
> |:----------:|:------:|:-----:|:----:|:----:|:----:|:----:|
> |      T     |   0.9  |  0.94 | 0.92 | 0.88 |  0.9 | 0.88 |
> |     T-O    |  0.18  |  0.16 | 0.26 | 0.22 | 0.18 |  0.2 |
>
> **Table R1:** Success rate of our method on the *BoxInBin* task when varying $\epsilon$.
>
>
> | $\tau$ | 0.0001 | 0.001 | 0.005 | 0.02 | 0.04 |   1  |
> |:------:|:------:|:-----:|:-----:|:----:|:----:|:----:|
> |    T   |  0.88  |  0.96 |  0.94 | 0.94 | 0.92 | 0.94 |
> |   T-O  |  0.28  |  0.26 |  0.2  | 0.28 | 0.22 | 0.22 |
>
> **Table R2:** Success rate of our method on the *BoxInBin* task when varying $\tau$.
>
> | winsize |    1 |    5 |   10 |   20 |   40 |  120 |
> |:-------:|-----:|-----:|-----:|-----:|-----:|-----:|
> |    T    | 0.82 | 0.92 | 0.86 |  0.9 | 0.94 | 0.84 |
> |   T-O   |  0.2 | 0.22 | 0.22 | 0.24 | 0.16 | 0.14 |
>
> **Table R3:** Success rate of our method on the *BoxInBin* task when varying the window size.
>
> | $\epsilon$ | 0.0001 | 0.001 | 0.01 | 0.04 |  0.1 |    1 |
> |:----------:|-------:|------:|-----:|-----:|-----:|-----:|
> |      T     |   0.84 |  0.88 |  0.8 | 0.76 | 0.78 | 0.76 |
> |     T-O    |    0.6 |  0.74 | 0.62 | 0.66 | 0.68 | 0.54 |
>
> **Table R4:** Success rate of our method on the *Lift* task when varying $\epsilon$.
>
> | $\tau$ | 0.0001 | 0.005 | 0.02 | 0.04 |  0.1 |    1 |
> |:------:|-------:|------:|-----:|-----:|-----:|-----:|
> |    T   |   0.78 |   0.7 | 0.76 | 0.76 | 0.78 | 0.74 |
> |   T-O  |   0.56 |  0.67 | 0.64 | 0.66 | 0.62 | 0.66 |
>
> **Table R5:** Success rate of our method on the *Lift* task when varying $\tau$.
>
> | winsize |    1 |   5 |   10 |   20 |   40 |  120 |
> |:-------:|-----:|----:|-----:|-----:|-----:|-----:|
> |    T    | 0.86 | 0.8 |  0.7 | 0.78 | 0.74 | 0.82 |
> |   T-O   | 0.66 | 0.6 | 0.67 | 0.58 | 0.56 |  0.6 |
>
> **Table R6:** Success rate of our method on the *Lift* task when varying the window size.
>
> - **Effectiveness with extremely limited target domain data:** We argue that in real-world robotic settings, **10–25 demonstrations constitute a highly limited amount of data—particularly when compared to prior works such as [1], which often require more than 100 demonstrations**. This limitation is further compounded in our evaluation, which involves diverse object poses, textures, shapes, and extended temporal horizons. To address the reviewer’s concern, we provide a comparative analysis of different methods under this low-data regime in the target domain, specifically for the *BoxInBin* task with the *Viewpoint3-Point* setting. The results show that our approach consistently outperforms the baselines, **demonstrating its robustness even with extremely limited supervision**.
>
> |           |     | Ours (CI)       | Co-train (CI)      | MMD (CI)        | T.-only (CI)    |
> |-----------|-----|-----------------|-----------------|-----------------|------------------|
> | **1 Demo**| T   | 0.56 (0.42, 0.70) | 0.46 (0.32, 0.60) | 0.42 (0.28, 0.56) | 0.00 (0.00, 0.00) |
> | **1 Demo**          | T-O | 0.28 (0.16, 0.40) | 0.00 (0.00, 0.00) | 0.16 (0.06, 0.26) | 0.00 (0.00, 0.00) |
> | **5 Demo**| T   | 0.70 (0.57, 0.83) | 0.38 (0.25, 0.51) | 0.34 (0.21, 0.47) | 0.46 (0.32, 0.60) |
> | **5 Demo**  | T-O | 0.32 (0.19, 0.45) | 0.22 (0.11, 0.33) | 0.22 (0.11, 0.33) | 0.00 (0.00, 0.00) |
>
> **Table R7:** Success rates and confidence intervals for various methods on the *BoxInBin* task with the *Viewpoint3-Point* setting, under scenarios where data from the target domain is extremely limited.
>
> - **Generalize to tasks with significant dynamics gaps:** As stated in Line 119-121, our work focuses on quasi-static tasks, which are representative of many everyday scenarios in home environments. In these settings, the observation gap is the primary bottleneck for sim-to-real transfer and presents a substantial challenge on its own. While handling dynamics mismatches is not the central focus of this work, **our method is built on a co-training framework, which inherently provides some robustness to dynamics gap**. Recent works[2, 3] have also demonstrated that co-training strategies can mitigate the effects of dynamics gaps and handle certain dynamic tasks. We acknowledge this as a limitation of the current study and view extending our framework to more dynamic tasks as a valuable direction for future research.
> - **Alternative domain alignment methods:** Our problem setting specifically targets sim-to-real transfer under out-of-distribution (OOD) conditions in the real world, with limited real-world demonstrations. The domain alignment methods mentioned by the reviewer[4, 5] primarily rely on adversarial training to learn a shared latent space for the transitions, but they **do not account for the significant data imbalance between the source (simulation) and target (real-world) domains—a core challenge in our setting**. While these approaches are relevant in broader domain adaptation contexts, they would require significant effort to be adapted to the problem formulation in this paper. We will cite and discuss these aspects in the related work section of the revised manuscript.
> - **Missing citations:** We will cite and discuss these works[4, 5, 6] in the revision.
> - **Dependence on demonstration alignment:** We would like to clarify the notion of "alignment" as used in our work. **Our method assumes alignment only at the level of task semantics**—that is, demonstrations across different domains are expected to operate on the same object geometry and pursue the same task objective. We **do not assume identical execution rates or strict temporal synchronization between trajectories**. To account for temporal variability, our framework incorporates a temporally aligned sampling strategy during training.
>     Comparable or even more restrictive assumptions regarding semantic alignment have been adopted in prior work. For example, Raychaudhuri et al.[4] referenced by the reviewer assumes that linearly normalized temporal positions correspond to the same task semantics—an assumption that may not hold in real-world robotics scenarios, where execution speeds can vary significantly across domains and task stages.
>     We contend that the semantic alignment assumption required by our method is **practical and readily achievable**, particularly given recent advances in perception systems[7], digital twin technologies[8, 9], and motion synthesis tools[10, 11] that enable the generation of diverse and semantically consistent demonstrations from a small number of source demonstrations.
>
> [1] Natural Language Can Help Bridge the Sim2Real Gap, RSS'24;
>
> [2] Empirical Analysis of Sim-and-Real Cotraining Of Diffusion Policies For Planar Pushing from Pixels, IROS'25;
>
> [3] Sim-and-Real Co-Training: A Simple Recipe for Vision-Based Robotic Manipulation, RSS'25;
>
> [4] Cross-domain Imitation from Observations, ICML'21;
>
> [5] Domain Adaptive Imitation Learning, ICML'20.
>
> [6] Dude: Dual Distribution-Aware Context Prompt Learning For Large Vision-Language Model, ACML'25
>
> [7] FoundationPose: Unified 6D Pose Estimation and Tracking of Novel Objects, CVPR'24;
>
> [8] Automated Creation of Digital Cousins for Robust Policy Learning, CoRL'24;
>
> [9] Reconciling Reality through Simulation: A Real-to-Sim-to-Real Approach for Robust Manipulation, RSS'24;
>
> [10] Mimicgen: A data generation system for scalable robot learning using human demonstrations, CoRL'23;
>
> [11] DemoGen: Synthetic Demonstration Generation for Data-Efficient Visuomotor Policy Learning, RSS'25.

---

> > ### Comment · Reviewer_PAX9 · 2025-08-02
> >
> > The authors adequately addressed all my concerns on sensitivity analysis and further scalability issues. Hence, I raised my score.

---

### Official Review · Reviewer_4KCC · 2025-06-23

**Clarity:** 3
**Significance:** 3
**Originality:** 2
**Rating:** 4
**Confidence:** 4

**Summary:**

This paper proposes a unified sim-and-real co-training framework for learning generalizable policies for robotic manipulation. It aims to address the challenges of costly real-world robot demonstration data acquisition and the sim-to-real domain gap by leveraging both a small number of real-world demonstrations and abundant simulated data. A key contribution is the proposal of an Optimal Transport (OT)-based loss function to align the joint distributions of observations and actions across domains, extended to an Unbalanced Optimal Transport (UOT) framework to handle imbalanced data. Furthermore, a temporally-aware sampling strategy is introduced to enhance the efficiency of mini-batch OT training. Through experiments on various robotic manipulation tasks, the proposed method intends to demonstrate improved real-world success rates and superior generalization capabilities, even for scenarios learned exclusively in simulation.

**Questions:**

1. Optimal Transport and Unbalanced Optimal Transport are well-known for domain adaptation. Please clearly explain what new insights or contributions, methodologically or theoretically, this paper offers compared to existing OT/UOT-based domain adaptation methodologies. Additional theoretical analysis is needed to demonstrate that this is not merely a combination of existing techniques.
2. To solidify the superiority of the proposed methodology, please conduct comparative experiments against at least three more recent and relevant sim-to-real or domain adaptation techniques. Furthermore, quantitatively analyze the method's robustness to real-world uncertainties (e.g., varying lighting conditions, object deformations, robot pose errors).
3. The paper claims to require "only a few real-world demonstrations." Please provide a detailed analysis of the minimum threshold for "a few real-world data" used, and how the diversity of this data (e.g., different angles, lighting, object instances) influences the results. Can you clearly demonstrate the method's effectiveness even with extremely limited real-world data (e.g., fewer than 5 demonstrations)?

**Ethical Concerns:**

["NO or VERY MINOR ethics concerns only"]

**Final Justification:**

The additional experiments and clarifications have addressed my initial concerns.

**Limitations:**

yes

**Paper Formatting Concerns:**

The paper generally follows the NeurIPS 2025 formatting guidelines and presents a clear structure. However, the spacing between the figures and their captions is too narrow, and the table captions should be positioned above the tables, not below.

**Quality:**

2

**Strengths And Weaknesses:**

### Strengths

- Addressing the sim-to-real transfer problem in robot learning remains a significant challenge, and the attempt to overcome the constraint of limited real-world data is relevant.
- The conceptual idea of applying Optimal Transport (OT) to align joint observation-action distributions, utilizing Unbalanced Optimal Transport (UOT) to manage data imbalance, and introducing a temporally aware sampling strategy is interesting.

### Weaknesses

- The proposed methodology appears to be primarily a combination of existing research on Optimal Transport and Unbalanced Optimal Transport for domain adaptation. There is a lack of clear explanation and sufficient theoretical justification for how this combination offers fundamental new theoretical or methodological contributions beyond existing methods. The temporally aware sampling appears to be an auxiliary technique for solving a specific problem rather than contributing core originality.
- While the presented experimental results show performance improvements in specific tasks, they lack rigor in several critical aspects.
    - Comparisons to other state-of-the-art and more robust sim-to-real or domain adaptation techniques are very limited. The current baselines are insufficient to adequately prove the superiority of the proposed method, which could lead to an overstatement of the paper's contributions.
    - While the paper emphasizes "a few real-world demonstrations," there is insufficient detailed information about the scale, diversity, and complexity of the actual datasets used. The OOD generalization results might be attributable to the specific dataset characteristics or the simplicity of the simulation environment rather than the intrinsic strength of the methodology. Validation across a wider range of domain gap scenarios is necessary.
- Unbalanced OT and the temporally-aware sampling can involve complex mathematical backgrounds, and their implementation and tuning in real systems are expected to be considerably complex. There is insufficient discussion on whether this added complexity sufficiently justifies the observed performance gains, and what potential issues might arise during real-world deployment.

---

> ### Author Rebuttal · Authors · 2025-07-31
>
> Thank you for your time and constructive feedback. We appreciate the positive recognition of our work in proposing a conceptually compelling framework for addressing challenging real-world robotics problems, particularly under the practical constraint of limited real-world demonstrations—a setting that is often overlooked in prior work. Below, we address the raised concerns in detail:
> - **Theoretical or methodological contributions:** We thank the reviewer for their feedback. **Our work addresses a critical emerging challenge in robot learning: how to achieve real-world generalization when training primarily on simulated data with only sparse real demonstrations.** While Optimal Transport (OT) has been used in domain adaptation, existing methods fail to handle the unique requirements of policy co-training—particularly the need to preserve action-relevant features across domains while coping with severe data imbalance ($N_{src}$ >> $N_{tgt}$) and partial state overlap. **Our key innovations include: (1) joint alignment of observations and actions (rather than marginal features alone), (2) an Unbalanced OT formulation to handle domain mismatch, and (3) temporally-aware sampling for sequential data—all specifically designed for the sim-to-real policy setting**.
>     The problem setting itself represents an important and understudied challenge. As large-scale simulation systems like MimicGen[1] enable massive synthetic datasets, the fundamental question becomes how to effectively leverage this data for real-world generalization, especially for states unseen in real demonstrations. Our framework provides the first systematic solution, achieving +30\% success rates on out-of-distribution real-world tasks compared to standard co-training (Table 2 and Table 4). This demonstrates that our methodological adaptations are not just applications of OT, but necessary innovations to bridge simulation and reality for policy learning.
>     We highlight the growing paradigm of scaling imitation learning by combining simulated and real-world data[2, 3]. Recent advances in robot data synthesis[1, 4, 5] enable controllable, large-scale simulation. **Our main contribution is to formalize and address the overlooked challenge of out-of-distribution (OOD) generalization in the target domain**. We propose a novel OT-based framework that addresses data imbalance and mini-batch limitations while retaining theoretical guarantees, and validate it through real-world experiments under practical constraints—often neglected in prior work.
> - **Complexity in implementation and tuning:** We thank the reviewer for pointing out the importance of evaluating the trade-off between algorithmic complexity and practical benefit. To address this concern, we conducted an additional ablation study to analyze the sensitivity of our method to its key hyperparameters, including the entropy regularization coefficient ($\epsilon$), the KL divergence penalty term ($\tau$), and the window size used in temporally aligned sampling. In each experiment, we varied a single hyperparameter while holding others fixed, trained the policy, and assessed performance via rollout evaluations. We report results for the *BoxInBin* task under the *Viewpoint-Image* setting and the *Lift* task under the *Texture-Image* setting, as shown in the following tables (Table R1-R6). The results show that **our method is robust to a range of reasonable hyperparameter choices** and does not require fine-grained tuning to achieve strong performance. Regarding the implementation complexity, while UOT and temporally-aware sampling do introduce some algorithmic overhead, **both components are modular and can be easily integrated into standard BC pipelines**. To further support reproducibility, we will release the codebase upon acceptance.
>
> | $\epsilon$ | 0.0001 | 0.001 | 0.01 | 0.04 |  0.1 |   1  |
> |:----------:|:------:|:-----:|:----:|:----:|:----:|:----:|
> |      T     |   0.9  |  0.94 | 0.92 | 0.88 |  0.9 | 0.88 |
> |     T-O    |  0.18  |  0.16 | 0.26 | 0.22 | 0.18 |  0.2 |
>
> **Table R1:** Success rate of our method on the *BoxInBin* task when varying $\epsilon$.
>
> | $\tau$ | 0.0001 | 0.001 | 0.005 | 0.02 | 0.04 |   1  |
> |:------:|:------:|:-----:|:-----:|:----:|:----:|:----:|
> |    T   |  0.88  |  0.96 |  0.94 | 0.94 | 0.92 | 0.94 |
> |   T-O  |  0.28  |  0.26 |  0.2  | 0.28 | 0.22 | 0.22 |
>
> **Table R2:** Success rate of our method on the *BoxInBin* task when varying $\tau$.
>
> | winsize |    1 |    5 |   10 |   20 |   40 |  120 |
> |:-------:|-----:|-----:|-----:|-----:|-----:|-----:|
> |    T    | 0.82 | 0.92 | 0.86 |  0.9 | 0.94 | 0.84 |
> |   T-O   |  0.2 | 0.22 | 0.22 | 0.24 | 0.16 | 0.14 |
>
> **Table R3:** Success rate of our method on the *BoxInBin* task when varying the window size.
>
> | $\epsilon$ | 0.0001 | 0.001 | 0.01 | 0.04 |  0.1 |    1 |
> |:----------:|-------:|------:|-----:|-----:|-----:|-----:|
> |      T     |   0.84 |  0.88 |  0.8 | 0.76 | 0.78 | 0.76 |
> |     T-O    |    0.6 |  0.74 | 0.62 | 0.66 | 0.68 | 0.54 |
>
> **Table R4:** Success rate of our method on the *Lift* task when varying $\epsilon$.
>
> | $\tau$ | 0.0001 | 0.005 | 0.02 | 0.04 |  0.1 |    1 |
> |:------:|-------:|------:|-----:|-----:|-----:|-----:|
> |    T   |   0.78 |   0.7 | 0.76 | 0.76 | 0.78 | 0.74 |
> |   T-O  |   0.56 |  0.67 | 0.64 | 0.66 | 0.62 | 0.66 |
>
> **Table R5:** Success rate of our method on the *Lift* task when varying $\tau$.
>
> | winsize |    1 |   5 |   10 |   20 |   40 |  120 |
> |:-------:|-----:|----:|-----:|-----:|-----:|-----:|
> |    T    | 0.86 | 0.8 |  0.7 | 0.78 | 0.74 | 0.82 |
> |   T-O   | 0.66 | 0.6 | 0.67 | 0.58 | 0.56 |  0.6 |
>
> **Table R6:** Success rate of our method on the *Lift* task when varying the window size.
> - **Effectiveness with extremely limited target domain data:** We argue that in real-world robotic settings, **10–25 demonstrations constitute a highly limited amount of data—particularly when compared to prior works such as [6], which often require more than 100 demonstrations**. This limitation is further compounded in our evaluation, which involves diverse object poses, textures, shapes, and extended temporal horizons. To address the reviewer’s concern, we provide a comparative analysis of different methods under this low-data regime in the target domain, specifically for the *BoxInBin* task with the *Viewpoint3-Point* setting. The results show that our approach consistently outperforms the baselines, **demonstrating its robustness even with extremely limited supervision**.
>
> |           |     | Ours (CI)       | Co-train (CI)      | MMD (CI)        | T.-only (CI)    |
> |-----------|-----|-----------------|-----------------|-----------------|------------------|
> | **1 Demo**| T   | 0.56 (0.42, 0.70) | 0.46 (0.32, 0.60) | 0.42 (0.28, 0.56) | 0.00 (0.00, 0.00) |
> | **1 Demo**          | T-O | 0.28 (0.16, 0.40) | 0.00 (0.00, 0.00) | 0.16 (0.06, 0.26) | 0.00 (0.00, 0.00) |
> | **5 Demo**| T   | 0.70 (0.57, 0.83) | 0.38 (0.25, 0.51) | 0.34 (0.21, 0.47) | 0.46 (0.32, 0.60) |
> | **5 Demo**  | T-O | 0.32 (0.19, 0.45) | 0.22 (0.11, 0.33) | 0.22 (0.11, 0.33) | 0.00 (0.00, 0.00) |
>
> **Table R7:** Success rates and confidence intervals for various methods on the *BoxInBin* task with the *Viewpoint3-Point* setting, under scenarios where data from the target domain is extremely limited.
>
> - **More recent and relevant sim-to-real baselines:** Our problem setting specifically focuses on **sim-to-real transfer under out-of-distribution (OOD) conditions** in the real world, with **limited access to real-world demonstrations** and **no prior knowledge of the target distribution**. While recent methods in robotic sim-to-real transfer have shown progress[2, 3, 7, 8, 9], **most do not directly tackle this challenging OOD generalization scenario**. For example, some approaches require human-in-the-loop interventions for online data collection after deployment[8], others rely on computationally expensive reinforcement learning pipelines[7], or adopt domain randomization[9], which assumes knowledge of the target domain distribution and may not generalize well when the test-time conditions fall far outside the training distribution. In contrast, our setting addresses scenarios where the target domain is not fully known a priori, and the task objective may vary or fall outside the training distribution, making traditional sim-to-real pipelines less suitable.
>     Among these works, co-training approaches[2, 3]—which train policies using a mixture of simulation and real-world data—are most aligned with our problem setting and have demonstrated strong performance. Therefore, we focus on these methods as baselines, as they are the **most relevant to our formulation and constraints, ensuring meaningful and fair comparisons**. We appreciate the reviewer for highlighting additional related works.
> - **Detailed information on real-world datasets:** We provide details of the real-world dataset in Section B.3 of the supplementary material. All baselines and our method are trained on the same dataset, ensuring a fair and consistent comparison.
>
> [1] Mimicgen: A data generation system for scalable robot learning using human demonstrations, CoRL'23;
>
> [2] Empirical Analysis of Sim-and-Real Cotraining Of Diffusion Policies For Planar Pushing from Pixels, IROS'25;
>
> [3] Sim-and-Real Co-Training: A Simple Recipe for Vision-Based Robotic Manipulation, RSS'25;
>
> [4] Dexmimicgen: Automated data generation for bimanual dexterous manipulation via imitation learning, ICRA'25;
>
> [5] DemoGen: Synthetic Demonstration Generation for Data-Efficient Visuomotor Policy Learning, RSS'25;
>
> [6] Natural Language Can Help Bridge the Sim2Real Gap, RSS'24;
>
> [7] From Imitation to Refinement -- Residual RL for Precise Assembly, arXiv'24;
>
> [8] TRANSIC: Sim-to-Real Policy Transfer by Learning from Online Correction, CoRL'24;
>
> [9] DeXtreme: Transfer of Agile In-hand Manipulation from Simulation to Reality, ICRA'23.

---

> > ### Comment · Reviewer_4KCC · 2025-08-02
> >
> > We appreciate the authors' rebuttal. However, several critical concerns remain fundamentally unresolved:
> >
> > **1. No substantive theoretical innovation beyond existing OT/UOT**
> >
> > The method primarily combines known techniques—joint observation-action alignment, unbalanced OT, and temporal sampling—without introducing new theoretical formulations or analytical insights. The rebuttal reiterates task-specific motivation but fails to demonstrate any principled advancement beyond prior OT/UOT-based domain adaptation. Without formal justification or novel derivations, the contribution remains an empirical application, not a methodological breakthrough.
> >
> > **2. Incomplete and selective empirical comparisons**
> >
> > The rebuttal does not provide comparisons to several established sim-to-real or domain adaptation baselines. Justifying their exclusion due to differing assumptions is unconvincing, particularly since many of these methods can still be adapted to the authors' setting. Without direct evaluation of stronger baselines, claims of superiority are not sufficiently supported.
> >
> > **3. Ambiguity around real-world generalization scope**
> >
> > While low-data experiments are added, the rebuttal does not clarify how variation in real-world conditions (e.g., lighting, object diversity, camera angle) impacts performance. Without controlled evaluation of such factors, the generalization claims remain speculative.
> >
> > The paper proposes a thoughtful integration of existing tools but lacks theoretical novelty and comprehensive empirical grounding. These core issues are not addressed in the rebuttal. Therefore, I remain my original evaluation.

---

> > > ### Author Response · Authors · 2025-08-05
> > >
> > > We sincerely thank the reviewer for their continued engagement and constructive critique. We would like to take this opportunity to clarify our contributions further, and address the key concerns raised regarding theoretical novelty, empirical comparison, and real-world generalization.
> > >
> > > 1. **On Theoretical Novelty**
> > >
> > > We acknowledge the reviewer's point that our work does not introduce new theorems for Optimal Transport. Our contribution is not a theoretical advance in OT itself, but rather a novel methodological framework that systematically adapts and integrates known techniques to solve a critical, and previously underexplored, problem in robot learning. We believe advances in this topic are of great interest to Robot Learning and the broader NeurIPS community.
> > >     Specifically, our method enables robust policy transfer from simulation to the real world when real-world data is extremely sparse—**a setting that diverges significantly from standard OT/UOT domain adaptation.** While the components of our method—**joint observation-action alignment, unbalanced OT, and temporal sampling**—have prior precedent individually, our key insight is in how they are **systematically integrated** to tackle this specific sim-to-real setting. Most prior OT/UOT-based methods focus on visual recognition tasks and do not address the **temporal and action-relevant structure**, or the **severe data imbalance** characteristic of real-world robotics.
> > >     In contrast, our formulation aligns trajectory segments in a space that explicitly preserves **policy-relevant temporal structure**. By combining **unbalanced OT across sequential data** with a tailored sampling strategy, our method distills transferable structure from simulated demonstrations to sparse real-world observations. To our knowledge, this trajectory-level unbalanced alignment has not been previously explored in this context.
> > >     While we do not claim a theoretical advance in OT itself, we believe that our **principled adaptation of OT for sim-to-real imitation learning**, and the novel use of unbalanced trajectory-level alignment, represent a **substantive methodological contribution**. We will make this framing clearer in the revision.
> > >
> > > 2. **On Empirical Comparisons**
> > >
> > > We thank the reviewer for raising this point. We chose baselines that are **most comparable in setting**, specifically co-training and domain adaptation methods that use both simulation and limited real data. As discussed in our previous response, many sim-to-real approaches assume access to dense real-world rewards, privileged information, or large quantities of real-world demonstrations, which are incompatible with our low-data, reward-free real-world setup. While adapting them is possible, doing so would require non-trivial redesigns that may obscure the core differences. **We would also be grateful if the reviewer could suggest specific papers they believe are adaptable to our low-data, reward-free, and offline problem formulation**. In the revised manuscript, we will expand our related work section to include a more detailed discussion of these other methods and explicitly clarify the differences in problem assumptions.
> > >
> > > 3. **Ambiguity Around Real-World Generalization**
> > >
> > > We appreciate the request for more clarity on real-world variation. Our OOD test set includes substantial variation in object pose, texture, shape, and camera viewpoint—**factors widely recognized as critical for policy generalization [1, 2, 3, 4, 5]**. These variations are explicitly detailed in **Appendix B.3** and visualized in **Figures 7 and 8** of our supplementary material. The strong performance of our method in these OOD settings, as shown in Table 2, is direct evidence of its generalization capabilities. We will add an additional statement in the main paper to highlight the scope of these real-world evaluations and point readers to the appendix for the full details.
> > >
> > >
> > >
> > > [1] What Matters in Learning from Large-Scale Datasets for Robot Manipulation, ICLR’25;
> > >
> > > [2] A Taxonomy for Evaluating Generalist Robot Policies, arXiv’25;
> > >
> > > [3] Decomposing the Generalization Gap in Imitation Learning for Visual Robotic Manipulation, ICRA’24;
> > >
> > > [4] The colosseum: A benchmark for evaluating generalization for robotic manipulation, RSS’24;
> > >
> > > [5] Efficient Data Collection for Robotic Manipulation via Compositional Generalization, RSS’24

---

> ### Author Response · Authors · 2025-08-07
>
> Thank you for your valuable comments. As the discussion deadline approaches, we would like to confirm if our latest response addresses all your concerns.

---

> > ### Comment · Area_Chair_yhTR · 2025-08-07
> >
> > Dear reviewer,
> >
> > Thanks for your initial reply. Please continue the discussion and let the authors know if they have (or have not) addressed your concerns. Also, please acknowledge the reply with the button for this.
> >
> > Best,
> > AC

---

> > ### Comment · Reviewer_4KCC · 2025-08-08
> >
> > I thank the authors for their clarifications. The positioning as a methodological adaptation for low-data sim-to-real transfer—with trajectory-level unbalanced OT preserving policy-relevant temporal structure—is now clearer. The focused baseline selection is reasonable given the problem constraints, and the appendix details confirm that key OOD variations are covered. While broader comparisons and controlled ablations would further strengthen the work, the rebuttal meaningfully addresses earlier concerns and improves the framing of the contribution. Therefore, I will raise my score accordingly.

---

> > > ### Author Response · Authors · 2025-08-08
> > >
> > > Thank you for the positive feedback and for your decision to raise our score. We are glad our clarifications helped address your concerns regarding the framing of our contribution and its evaluation. We appreciate your constructive engagement throughout the review process.

---

### Official Review · Reviewer_aDmq · 2025-06-28

**Clarity:** 2
**Significance:** 3
**Originality:** 3
**Rating:** 5
**Confidence:** 4

**Summary:**

This submission proposes an algorithm for sim and real co-training of robotic manipulation skills using Temporally Aligned Sampling and Unbalanced Optimal Transport. It is assumed that we have access to a large sim training dataset and a small real-world training dataset. The goal is to generalize to novel states in the real-world that are covered in the sim dataset but not in the real-world dataset. This is achieved by training an image or point cloud encoder with a regularization loss computed using optimal transport. It is shown that performing behavior cloning with this encoder leads to positive sim-to-real and sim-to-sim transfer and the learned policies generalize to states previously unseen in the target domain. The proposed method outperforms baselines that perform co-training on sim and real data and a simpler version of visual feature alignment between sim and real. The method is tested in a real-world robot experiment with a Franka arm.

**Questions:**

I do not fully understand the motivation in Section 4.3. It is stated that naively sampling pairs of transitions from the source and the target dataset “may yield pairs from different stages of tasks”. However, I thought that the optimal transport was computed over the entire episodes; wouldn’t the OT computation already take care of aligning different stages of the task across different trajectories?

**Ethical Concerns:**

["NO or VERY MINOR ethics concerns only"]

**Final Justification:**

The submission proposes a novel method for co-training on a large out-of-domain dataset and a small in-domain dataset. The paper contains strong experiments in both simulation and in the real world. Further, the authors performed an additional hyper-parameter sensitivity analysis for the rebuttal, which further increases my confidence in their method.

**Limitations:**

The limitations of the method are addressed in the conclusion. In particular, the section mentions lack of sim to real dynamics alignment and limited dexterity of skills generated by MimicGen-style demonstration cloning.

**Paper Formatting Concerns:**

None.

**Quality:**

3

**Strengths And Weaknesses:**

## Strengths

**Novel combination of known methods.** While sim and real co-training, unbalanced optimal transport and temporally aligned sampling are known methods, their combination to solve visual sim to real alignment that is done in this submission is novel. In particular, unbalanced optimal transport is useful because not every state in the source domain might have a corresponding state in the target domain.

**Excellent video.** The video provided in the supplementary material clearly communicates the idea that we are moving from a wide reset distribution in sim to a narrow reset distribution in real-world demonstrations and then shows examples of real-world policy inference with an OOD reset distribution. I think this is a great showcase of the idea of learning joint sim and real embeddings using temporally-aligned optimal transport.

**Evaluation with both images and point clouds.** The submission includes evaluation of diffusion policies with image encoders and 3D diffusion policies with point cloud encoders. Both evaluations highlight the usefulness of optimal transport in sim and real co-training. Using both images and point clouds increases the impact this paper could have in the robot learning community.

**Evaluation of OOD transfer.** The experiments specifically test transfer to OOD states but in the sim and in the real. Showing positive transfer in this setting is a step towards learning generalist real-world manipulation policies.

## Weaknesses

**No strong real-world baseline / oracle.** The performance of all methods in the real-world robot experiments is relatively weak. For example, all methods are limited to a 40% success rate for the task of stacking two blocks. This is not entirely unexpected, as the methods are provided with only 25 real-world demonstrations and the distribution shifts between training and testing. This set of experiments is missing an oracle that provides an upper bound on the real-world robot performance. Since we do not know how well a diffusion policy would perform with a few hundred demonstrations, it is not clear if the low performance is due to a lack of data or other problems, such as incorrect controller gains or bugs in real-world data collection. Providing an oracle performance would better contextualize the real-world results.

**Unclear if method could be reproduced from its description.** The method section contains frequent references to prior works for practical algorithms and an algorithmic description of the full method is delegated to Appendix C, Algorithm 1. Even Algorithm 1 lacks details needed to fully understand the computation done during training. I would suggest adding more details into Algorithm 1 and providing more guidance for reproducibility.

**Unclear sensitivity to hyper-parameters.** Each training step includes temporal alignment computation and optimal transport computation. It is unclear how the hyper-parameters of these two steps are tuned and how sensitive the method is to small changes in these hyper-parameters. This could be clarified in an additional ablation study.

---

> ### Author Rebuttal · Authors · 2025-07-31
>
> Thank you for your time and thoughtful feedback. We are grateful for the positive recognition of the novelty of our work, the high-quality accompanying video, and the thorough evaluations conducted across diverse modalities and domains, particularly under challenging out-of-distribution (OOD) conditions. Below, we address the raised concerns in detail:
> - **No strong real-world baseline/oracle:** We interpret the reviewer’s concern regarding the “40\% success rate for the real-world Stack task” as referring to the results in Table 2, which correspond to the out-of-distribution (OOD) generalization setting—specifically, cube placements in novel poses not seen during training. **These OOD configurations are intentionally challenging and designed to evaluate a model’s ability to generalize beyond observed real-world data**.
>     To provide additional context, we also report in-distribution (ID) performance for all methods, where our approach achieves a 90\% success rate (see Table 6 in the supplementary). This sharp contrast between ID and OOD performance **confirms the validity of our hardware setup and data collection process, while underscoring the core challenge of sim-to-real generalization under OOD conditions—the primary focus of our work**.
>     Regarding the absence of a strong real-world oracle, we emphasize that our work is intentionally situated in the low-data regime. This reflects the practical difficulty of collecting large numbers of real-world demonstrations across diverse OOD scenarios. For example, training a diffusion policy solely on real-world data with sufficient OOD coverage would typically require hundreds of demonstrations[1]—an expensive and often impractical requirement.
>     **Our setting is deliberately designed to reflect these real-world constraints**. Instead of relying on extensive real-world data, we explore how simulation data can be effectively leveraged—combined with limited real-world supervision—to bridge the sim-to-real gap under OOD target distributions.
> - **Reproducibility:** To enhance the reproducibility of our work, we plan to include code examples illustrating the implementation of the core components in the revised version. We also intend to release the codebase and dataset upon acceptance.
> - **Sensitivity to hyper-parameters:** We conducted an ablation study to evaluate the sensitivity of our method to key hyperparameters, including the entropy regularization coefficient ($\epsilon$), the KL divergence penalty term ($\tau$), and the window size used in temporally aligned sampling. In each experiment, we varied a single hyperparameter while keeping the others fixed, trained the policy, and assessed its performance via rollouts. We report results for the *BoxInBin* task under the *Viewpoint-Image* setting and the *Lift* task under the *Texture-Image* setting, as shown in the following tables (Table R1-R6). The results indicate that **our method is robust to hyperparameter variations** within reasonable ranges. Specifically, performance remains stable when $\epsilon$ and $\tau$ are set between 0.001 and 0.1, and when the window size is varied between 5 and 20. Our method consistently outperforms the co-training baseline in out-of-distribution (OOD) scenarios, where the baseline achieves a success rate of 0.14 on *BoxInBin* and 0.6 on *Lift*. These findings suggest that, although our approach introduces additional components, it does not require extensive tuning and offers clear advantages in terms of generalization.
>
> | $\epsilon$ | 0.0001 | 0.001 | 0.01 | 0.04 |  0.1 |   1  |
> |:----------:|:------:|:-----:|:----:|:----:|:----:|:----:|
> |      T     |   0.9  |  0.94 | 0.92 | 0.88 |  0.9 | 0.88 |
> |     T-O    |  0.18  |  0.16 | 0.26 | 0.22 | 0.18 |  0.2 |
>
> **Table R1:** Success rate of our method on the *BoxInBin* task when varying $\epsilon$.
>
>
> | $\tau$ | 0.0001 | 0.001 | 0.005 | 0.02 | 0.04 |   1  |
> |:------:|:------:|:-----:|:-----:|:----:|:----:|:----:|
> |    T   |  0.88  |  0.96 |  0.94 | 0.94 | 0.92 | 0.94 |
> |   T-O  |  0.28  |  0.26 |  0.2  | 0.28 | 0.22 | 0.22 |
>
> **Table R2:** Success rate of our method on the *BoxInBin* task when varying $\tau$.
>
> | winsize |    1 |    5 |   10 |   20 |   40 |  120 |
> |:-------:|-----:|-----:|-----:|-----:|-----:|-----:|
> |    T    | 0.82 | 0.92 | 0.86 |  0.9 | 0.94 | 0.84 |
> |   T-O   |  0.2 | 0.22 | 0.22 | 0.24 | 0.16 | 0.14 |
>
> **Table R3:** Success rate of our method on the *BoxInBin* task when varying the window size.
>
> | $\epsilon$ | 0.0001 | 0.001 | 0.01 | 0.04 |  0.1 |    1 |
> |:----------:|-------:|------:|-----:|-----:|-----:|-----:|
> |      T     |   0.84 |  0.88 |  0.8 | 0.76 | 0.78 | 0.76 |
> |     T-O    |    0.6 |  0.74 | 0.62 | 0.66 | 0.68 | 0.54 |
>
> **Table R4:** Success rate of our method on the *Lift* task when varying $\epsilon$.
>
> | $\tau$ | 0.0001 | 0.005 | 0.02 | 0.04 |  0.1 |    1 |
> |:------:|-------:|------:|-----:|-----:|-----:|-----:|
> |    T   |   0.78 |   0.7 | 0.76 | 0.76 | 0.78 | 0.74 |
> |   T-O  |   0.56 |  0.67 | 0.64 | 0.66 | 0.62 | 0.66 |
>
> **Table R5:** Success rate of our method on the *Lift* task when varying $\tau$.
>
> | winsize |    1 |   5 |   10 |   20 |   40 |  120 |
> |:-------:|-----:|----:|-----:|-----:|-----:|-----:|
> |    T    | 0.86 | 0.8 |  0.7 | 0.78 | 0.74 | 0.82 |
> |   T-O   | 0.66 | 0.6 | 0.67 | 0.58 | 0.56 |  0.6 |
>
> **Table R6:** Success rate of our method on the *Lift* task when varying the window size.
>
> - **The importance of sampling strategy:** Although Optimal Transport (OT) can theoretically align samples across different task stages when full episodes are used, this approach is often impractical due to computation constraints. In mini-batch training with limited batch sizes, especially for multi-stage or long-horizon tasks, **semantically aligned samples may rarely co-occur within the same batch. Our sampling strategy addresses this by increasing the likelihood of including aligned pairs, thereby improving alignment quality and accelerating convergence**. As demonstrated in Figure 9, this approach leads to significant improvements in policy performance.
>
> [1] DROID: A Large-Scale In-the-Wild Robot Manipulation Dataset, RSS'24.

---

> > ### Comment · Reviewer_aDmq · 2025-08-05
> > **Response**
> >
> > Thank you for the rebuttal and the additional experiment. Your exploration of hyper-parameters gives me greater confidence in the significance of your results. I am in favor of accepting this paper and I will raise my score.

---

### Official Review · Reviewer_kja7 · 2025-06-29

**Clarity:** 2
**Significance:** 3
**Originality:** 3
**Rating:** 4
**Confidence:** 4

**Summary:**

This paper deals with the problem is using simulator data to train a behavioral cloned policy that can be transferred into the real world or a separate target environment which might have significant differences in perceptual space compared to the source simulator domain. It addresses this problem by assuming some data can be collected in the target domain with expert demonstration. Then it uses the idea of co-training on source and target data to train a policy that can leverage the large amount of data in the source and the grounded data in the target domain.

The contribution of this paper is to add in an unbalanced optimal transport loss that will pull together the representations for observations in the source and target domains that are behaviorally similar (based on  observation, action pairs) while ensuring that the representations of observations that are not close to these few demonstrations don't collapse due to the unbalanced data volumes.

Evaluation is done in both sim-to-sim as well as sim-to-real settings, along with checking if adding more simulation data helps this approach learn better policies. t-SNE plots are also used to validate if the learned representations are close to each other.

**Questions:**

* Main questions are about validity of results in paper. Those are clarified in the weaknesses section above. Sufficient answers to them would be good enough for me to recommend acceptance.
* Assuming the dynamics of the environment is the same in source and target, could $ f(o^i\_{src}), f(o^{i+1}\_{src})$ be compared to $f(o^j_{tgt}), f(o^{j+1}_{tgt})$ (i.e. transitions instead of state action pairs)to get a similar objective to the one proposed in the paper? Or does the action/proprioception act as an oracle that can be used by the objective to decide which observations to align?

**Ethical Concerns:**

["NO or VERY MINOR ethics concerns only"]

**Final Justification:**

The additional results, with the information on number of trials as well as the variance in the results, have given me the confidence to increase my score.

**Limitations:**

Yes

**Quality:**

3

**Strengths And Weaknesses:**

## Strengths
* The paper has a clear aim, a clear contribution, and experiments for the most part are designed to validate that contribution
* Section 3 describing the technique is fairly well written and should not leave an intelligent reader with too many doubts. Appropriate references are made for a diligent reader to follow up and understand details.
* Practical implementation details are called out in Section 3 instead of hidden in the appendix, which is appreciated.
* Figures are clear
* Most relevant work is cited. See some caveats in weaknesses

## Weaknesses
* The main reason for my on-the-fence recommendation is that this paper depends a lot on empirical evaluation, and there doesn't seem to be any statistical guarantee on any of the results in the paper. None of the tables discusses the number of trials, variance in the results, confidence bounds, or statistical significance in the results. Values are made bold in tables without clarifying what the criterion is for this formatting; are these values the highest numbers, or are they significantly higher? While I forgive forgoing these for real robot experiments, I would require some statistical guarantee from the sim-to-sim experiments to approve this paper. Below I now go in order from the greatest weaknesses to nitpicks.
* The hypotheses laid out at the beginning of Section 5 are not the same ones mentioned later on page 8. Same or similar wording when mentioning the hypotheses being validated will help a reader to ground the experiments.
* The phrasing of the hypotheses at the beginning of Section 5 is not clear enough: when mentioning words denoting comparison such as "improves" and "enables generalization", it is useful to point out what it is improving over (I am assuming baseline co-training performance) or generalizing better than (I am assuming other baselines).
* `Our method is applicable to different observation modalities` is vague. It is better to say `multiple observation modalities`
* On line 297, the paper points to the appendix for the validation result of the first hypothesis. First, please reference the particular figure or table a reader should reference. Second, if an result is essential to validate the presented approach, it should be part of the main paper and not the appendix.
* On line 314, the paper says `our approach effectively learns domain-invariant features that capture task-relevant information regardless of the sensory input.` This seems like over-claiming. Better to say that it does so `on multiple sensory modalities`.
* In the paragraph starting line 315, results are talked about without referencing the actual table being referenced.
* In line 336, Figure 5 is no longer the correct figure being referenced (It is again in the appendix, and is now Figure 10).
* Line 126, I am assuming $S_{src}$ is the number of states covered in the source domain. This distinction from the source state space $\mathcal{S}$ should be better clarified.
* Emission functions used in line 113 should be defined earlier when defining observation spaces.
* I understand that overcoming dynamics mismatch is not an aim of this paper, but related work should acknowledge that dynamics mismatch is one of the challenges in sim-to-real transfer, but clarify that this paper does not deal with it. The authors should reference some seminal papers in this approach. I refrain from suggesting any to avoid biasing the authors.
* Line 156, ` ... while preserve detailed information ...` should be `... while preserving detailed information ...`

---

> ### Author Rebuttal · Authors · 2025-07-31
>
> Thank you for your time and constructive feedback. We sincerely appreciate the positive recognition of our work’s clear motivations and contributions, the sufficient implementation details provided, and the well-designed experiments validating these contributions. Below, we address each of the concerns in detail:
> - **Statistical guarantee from the sim-to-sim experiments:** For each task and method, we perform **50 policy rollouts with randomly sampled initial states, following standard protocol in prior works [1, 2, 3]**, where the number of trials is generally considered sufficient to ensure statistical significance. We additionally **report 95\% confidence intervals using standard error** in Table R1 and Table R2, with the criterion used in prior works[4], for all simulation experiments. In our tables, bolded numbers indicate the best-performing method for each task.
> | method    | BoxInBin (V3) T | BoxInBin (V3) T‑O | BoxInBin \(P\) T | BoxInBin \(P\) T‑O | Lift (V1) T | Lift (V1) T‑O | Stack (V1) T | Stack (V1) T‑O | Square (V1) T | Square (V1) T‑O | MugHang (V1) T | MugHang (V1) T‑O | Average T | Average T‑O |
> |-----------|---------------|---------------|--------------|--------------|--------------|--------------|---------------|---------------|---------------|---------------|----------------|----------------|--------------|--------------|
> | `Ours`    | 0.65 (0.51, 0.77) | 0.04 (−0.01, 0.09) | 0.86 (0.76, 0.96) | 0.02 (−0.02, 0.06) | 0.88 (0.79, 0.97) | 0.26 (0.14, 0.38) | 0.66 (0.53, 0.79) | 0.52 (0.38, 0.66) | 0.68 (0.55, 0.81) | 0.54 (0.40, 0.68) | 0.96 (0.91, 1.01) | 0.82 (0.71, 0.93) | 0.78 (0.73, 0.83) | 0.36 (0.31, 0.41) |
> | `Co-train` | 0.44 (0.30, 0.58) | 0.04 (−0.01, 0.09) | 0.76 (0.64, 0.88) | 0.00 (0.00, 0.00) | 0.90 (0.82, 0.98) | 0.14 (0.04, 0.24) | 0.54 (0.40, 0.68) | 0.34 (0.21, 0.47) | 0.66 (0.53, 0.79) | 0.46 (0.32, 0.60) | 0.98 (0.94, 1.02) | 0.72 (0.60, 0.84) | 0.71 (0.66, 0.76) | 0.28 (0.23, 0.33) |
> | `MMD`     | 0.38 (0.25, 0.51) | 0.00 (0.00, 0.00) | 0.18 (0.07, 0.29) | 0.04 (−0.01, 0.09) | 0.82 (0.71, 0.93) | 0.16 (0.06, 0.26) | 0.44 (0.30, 0.58) | 0.40 (0.26, 0.54) | 0.38 (0.25, 0.51) | 0.34 (0.21, 0.47) | 0.80 (0.69, 0.91) | 0.70 (0.57, 0.83) | 0.50 (0.44, 0.56) | 0.30 (0.24, 0.35) |
> | `T.-only`     | 0.30 (0.17, 0.43) | 0.00 (0.00, 0.00) | 0.20 (0.09, 0.31) | 0.00 (0.00, 0.00) | 0.82 (0.71, 0.93) | 0.00 (0.00, 0.00) | 0.42 (0.28, 0.56) | 0.00 (0.00, 0.00) | 0.48 (0.34, 0.62) | 0.00 (0.00, 0.00) | 0.64 (0.51, 0.77) | 0.00 (0.00, 0.00) | 0.48 (0.42, 0.54) | 0.00 (0.00, 0.00) |
> | `S.-only`     | 0.00 (0.00, 0.00) | 0.00 (0.00, 0.00) | 0.00 (0.00, 0.00) | 0.00 (0.00, 0.00) | 0.00 (0.00, 0.00) | 0.00 (0.00, 0.00) | 0.00 (0.00, 0.00) | 0.00 (0.00, 0.00) | 0.00 (0.00, 0.00) | 0.00 (0.00, 0.00) | 0.00 (0.00, 0.00) | 0.00 (0.00, 0.00) | 0.00 (0.00, 0.00) | 0.00 (0.00, 0.00) |
>
> **Table R1:** Sim-to-sim success rates with confidence intervals for image-based policies.
>
> | method     | BoxInBin (V3) T | BoxInBin (V3) T‑O | BoxInBin \(P\) T | BoxInBin \(P\) T‑O | Lift (V1) T | Lift (V1) T‑O | Stack (V1) T | Stack (V1) T‑O | Square (V1) T | Square (V1) T‑O | MugHang (V1) T | MugHang (V1) T‑O | Average T | Average T‑O |
> |------------|---------------|---------------|--------------|--------------|--------------|--------------|---------------|---------------|---------------|---------------|----------------|----------------|---------------|---------------|
> | `Ours`     | 0.84 (0.74, 0.94) | 0.58 (0.42, 0.70) | 0.80 (0.69, 0.91) | 0.76 (0.64, 0.88) | 0.80 (0.69, 0.91) | 0.60 (0.46, 0.74) | 0.82 (0.71, 0.93) | 0.86 (0.76, 0.96) | 0.42 (0.28, 0.56) | 0.38 (0.25, 0.51) | 0.40 (0.26, 0.54) | 0.34 (0.21, 0.47) | 0.68 (0.63, 0.73) | 0.59 (0.53, 0.65) |
> | `Co-train`  | 0.76 (0.64, 0.88) | 0.52 (0.38, 0.66) | 0.70 (0.57, 0.83) | 0.66 (0.53, 0.79) | 0.92 (0.84, 1.00) | 0.48 (0.34, 0.62) | 0.86 (0.76, 0.96) | 0.72 (0.60, 0.84) | 0.24 (0.12, 0.36) | 0.24 (0.12, 0.36) | 0.26 (0.14, 0.38) | 0.22 (0.11, 0.33) | 0.62 (0.57, 0.67) | 0.47 (0.41, 0.53) |
> | `MMD`      | 0.50 (0.36, 0.64) | 0.38 (0.25, 0.51) | 0.66 (0.53, 0.79) | 0.50 (0.36, 0.64) | 0.56 (0.42, 0.70) | 0.52 (0.38, 0.66) | 0.70 (0.57, 0.83) | 0.66 (0.53, 0.79) | 0.18 (0.07, 0.29) | 0.12 (0.03, 0.21) | 0.18 (0.07, 0.29) | 0.20 (0.09, 0.31) | 0.46 (0.40, 0.52) | 0.40 (0.34, 0.46) |
> | `T.-only`      | 0.42 (0.28, 0.56) | 0.00 (0.00, 0.00) | 0.58 (0.42, 0.70) | 0.00 (0.00, 0.00) | 0.60 (0.46, 0.74) | 0.00 (0.00, 0.00) | 0.32 (0.19, 0.45) | 0.00 (0.00, 0.00) | 0.16 (0.06, 0.26) | 0.00 (0.00, 0.00) | 0.18 (0.07, 0.29) | 0.00 (0.00, 0.00) | 0.38 (0.33, 0.43) | 0.00 (0.00, 0.00) |
> | `S.-only`      | 0.08 (0.00, 0.16) | 0.10 (0.02, 0.18) | 0.52 (0.38, 0.66) | 0.60 (0.46, 0.74) | 0.32 (0.19, 0.45) | 0.40 (0.26, 0.54) | 0.52 (0.38, 0.66) | 0.64 (0.51, 0.77) | 0.10 (0.02, 0.18) | 0.08 (0.00, 0.16) | 0.12 (0.03, 0.21) | 0.10 (0.02, 0.18) | 0.28 (0.23, 0.33) | 0.32 (0.27, 0.37) |
>
> **Table R2:** Sim-to-sim success rates with confidence intervals for point cloud-based policies.
>
> - **Transferring transitions instead of state-action pairs:** Our proposed framework **does not assume a fixed temporal length for input observations**, so extending it to incorporate temporal information—such as full transitions—is straightforward. In this work, we adopt standard behavioral cloning (BC) architectures, namely Diffusion Policy[5] and 3D Diffusion Policy[6], as concrete instantiations to validate the core function of our framework. Exploring extensions that leverage transition-level alignment is a promising direction for future work.
> - **Inconsistent wording in hypotheses:** We will revise the wording of the hypotheses to align more closely with the conclusions presented in the experimental section.
> - **Missing/incorrect reference:** We will correct all reference issues related to tables and figures in the revision.
> - **Unclear/inappropriate wording:** We will carefully review and address any ambiguous or awkward phrasing in the revision.
>
> [1] What Matters in Learning from Offline Human Demonstrations for Robot Manipulation, CoRL'21;
>
> [2] LIBERO: Benchmarking Knowledge Transfer for Lifelong Robot Learning, NeurIPS'23
>
> [3] What Matters in Learning from Large-Scale Datasets for Robot Manipulation, ICLR'25;
>
> [4] Empirical Analysis of Sim-and-Real Cotraining Of Diffusion Policies For Planar Pushing from Pixels, IROS'25.
>
> [5] Diffusion Policy: Visuomotor Policy Learning via Action Diffusion, RSS'23;
>
> [6] 3D Diffusion Policy: Generalizable Visuomotor Policy Learning via Simple 3D Representations, RSS'24.

---

> > ### Comment · Reviewer_kja7 · 2025-08-05
> > **Response to Author Rebuttal**
> >
> > I thank the authors for providing more details about the experimental evaluation. While the proposed method doesn't seem to be statistically better than the co-train baseline the performance is better than this baseline on almost all evaluations and the 95% confidence intervals have low overlap. If the authors could perform a statistical test to validate the significance of the difference it would be ideal.
> >
> > I thank the authors for the rest of their responses. I will raise my score slightly, since it is impossible for me to verify the changes the authors have agreed to make to the wording of the paper.

---

> ### Author Response · Authors · 2025-08-05
>
> Dear reviewer kja7,
>
> We are writing to kindly follow up regarding our response to the reviewer comments for our submission. Please let us know if any further clarification is needed. We appreciate your time and consideration!

---

### Note · Authors · 2025-08-13

We thank the reviewers and the AC for their thoughtful feedback and productive discussion. In our rebuttal, we addressed all raised concerns: we clarified the core technical contributions and their positioning; specified and justified the evaluation protocol and baseline comparisons; and provided analyses on sensitivity and robustness of the full system.

We appreciate the recognition of our paper’s motivation and contributions. Specifically, we formulate an overlooked yet important setting: robotic low-data sim-to-real transfer with an explicit focus on out-of-distribution (OOD) conditions, which differ substantially from the assumptions in standard domain adaptation. We present a principled framework to systematically tackle these challenges and evaluate it comprehensively across diverse modalities and domains, including challenging OOD scenarios.

We are pleased that the reviewers have found the clarifications and additional experiments provided during the rebuttal process to be satisfactory and are unanimously positive about our contributions. We look forward to refining the presentation and extending the evaluation in future work.

---

### Decision · Program_Chairs · 2025-09-17

**Decision:**

Accept (poster)

**Comment:**

First concern is lack of statistical analysis; response here seems to address policy evaluation; inspired by theory but loses that in the approach; all reviewers positive

This paper studies the problem of using simulators to provide data for behavior cloning. One critical challenge is the sim2real visual gap and the work addresses this by learning a shared feature space between sim and real. Experiments show the approach enables sim2real transfer of manipulation policies.

Reviewers generally liked the papers motivation, clear contribution, and experimental design. A number of points about the writing were highly praised. The aim to use simulation to provide demonstration data is well-motivated and the approach is sound and novel.

One of the main concerns was a lack of statistical analysis given in the paper. This was clearly addressed in the rebuttal. Another limitation is that the paper only addresses visual sim2real gap and not dynamics sim2real gap. This limitation should be clearly acknowledged in the paper.